# Direct detection of coupled proton and electron transfers in human manganese superoxide dismutase

Jahaun Azadmanesh [1], William E. Lutz [2], Leighton Coates [3], Kevin L. Weiss [4] & Gloria E. O. Borgstahl [1,2 ✉]

Human manganese superoxide dismutase is a critical oxidoreductase found in the mitochondrial matrix. Concerted proton and electron transfers are used by the enzyme to rid the mitochondria of $O_2^{\bullet-}$. The mechanisms of concerted transfer enzymes are typically unknown due to the difficulties in detecting the protonation states of specific residues and solvent molecules at particular redox states. Here, neutron diffraction of two redox-controlled manganese superoxide dismutase crystals reveal the all-atom structures of $Mn^{3+}$ and $Mn^{2+}$ enzyme forms. The structures deliver direct data on protonation changes between oxidation states of the metal. Observations include glutamine deprotonation, the involvement of tyrosine and histidine with altered $pK_a$s, and four unusual strong-short hydrogen bonds, including a low barrier hydrogen bond. We report a concerted proton and electron transfer mechanism for human manganese superoxide dismutase from the direct visualization of active site protons in $Mn^{3+}$ and $Mn^{2+}$ redox states.

[1] Department of Biochemistry and Molecular Biology, 985870 Nebraska Medical Center, Omaha, NE, USA. [2] Eppley Institute for Cancer and Allied Diseases, 986805 Nebraska Medical Center, Omaha, NE, USA. [3] Second Target Station, Oak Ridge National Laboratory, Oak Ridge, TN, USA. [4] Neutron Scattering Division, Oak Ridge National Laboratory, Oak Ridge, TN, USA. ✉email: gborgstahl@unmc.edu

Oxidoreductases are a large class of enzymes that use unpaired electrons to facilitate redox reactions with other chemical species and are involved in nearly all aspects of life. Oxidoreductase electron transfers are almost always coupled with a proton transfer[1]. Concerted proton and electron transfer (CPET) permits a thermodynamically favorable redox reaction, is efficient, and is an integral part of enzymes with the fastest catalytic rates[2–4]. Particularly noteworthy is the prominence of CPET enzymes that regulate the concentration of reactive oxygen species (ROS) in the cell. ROS levels are central to programmed cell death and abnormal regulation by these oxidoreductases play significant roles in cancer and cardiovascular diseases[5]. CPETs are therefore of significant interest to study but a mechanistic understanding of these enzymes is still lacking. Deciphering these fundamental biochemical reactions is not only significant for its role in diseases, but for the biomedical design of CPET-dependent therapeutic interventions, irradiation protectants, and electrochemical biosensors[6,7].

Human manganese superoxide dismutase (MnSOD) is a CPET-based oxidoreductase found in the mitochondrial matrix that reduces ROS levels by eliminating $O_2^{•-}$ with the unpaired electrons of the active site metal. The Mn ion is coordinated to inner-sphere residues His26, His74, His163, Asp159, and a single-oxygen species that could be either $H_2O$ or $^-OH$ (designated WAT1, Fig. 1). Trivalent Mn oxidizes $O_2^{•-}$ to $O_2$ ($k_1 = 1.5\ nM^{-1}\ s^{-1}$)[8] and the resulting divalent Mn reduces another $O_2^{•-}$ molecule to $H_2O_2$ ($k_2 = 1.1\ nM^{-1}\ s^{-1}$)[8]. This is the only means the mitochondrial matrix has to keep $O_2^{•-}$ levels low enough to avoid damage to macromolecules and destruction of cellular function[9].

$$(k_1)\quad Mn^{3+} + O_2^{•-} \leftrightarrow Mn^{2+} + O_2$$
$$(k_2)\quad Mn^{2+} + O_2^{•-} + 2H^+ \leftrightarrow Mn^{3+} + H_2O_2$$

The major endogenous source of $O_2^{•-}$ is from electrons inadvertently leaking from the electron transport chain. Dysfunctional MnSOD activity, therefore, poses significant consequences on the mitochondria that contributes to several diseases. Genetic aberrations of MnSOD are associated with several cancer types, with mammary and prostate cancers being the most frequently noted in curated databases[10]. Polymorphisms of MnSOD have also been noted to be a predictor for deficient vascular function[11]. Therefore, the ability for MnSOD to utilize the high reaction rate and efficiency ($k_{cat}/K_m > \sim 10^9\ M^{-1}\ s^{-1}$) of its CPET mechanism is correlated with the preservation of health[8].

The CPET mechanism of MnSOD and the majority of other oxidoreductases has yet to be defined at the atomic level. The limitation in studying CPETs is the difficulty in directly detecting the protonation states of ionizable residues, solvent/ligands at the active site, and correlating them with the electronic state of the active site metal. The second sphere of MnSOD harbors five residues (His30, Tyr34, Gln143, Glu162, and Tyr166 (Fig. 1)) and information about their protonation states would be of significant value in deciphering a catalytic mechanism. X-ray and spectroscopic techniques have been unable to provide this information due to the poor scattering of hydrogen atoms and the difficulty in discerning spectra for specific titratable positions.

Current mechanistic knowledge has relied on X-ray crystal structures and kinetic studies of point mutants to infer the role of second-sphere residues. Mutations of Gln143, Tyr34, or His30 do not significantly alter the active site structure although they are detrimental to the kinetic rates of the enzyme. The Gln143Asn point mutation leads to near ablation of catalysis, Tyr34Phe depletes activity only for the $Mn^{2+} \rightarrow Mn^{3+}$ redox transition, and His30Gln has 40 and 82% of wild-type activity for the $Mn^{3+} \rightarrow Mn^{2+}$ and $Mn^{2+} \rightarrow Mn^{3+}$ transitions, respectively[8,12–14]. Quantum mechanic and molecular mechanic (QM/MM) calculations have been applied to X-ray structures for insight into the possible proton transfers. They have suggested protonation of WAT1 from $^-OH$ to $H_2O$ during the $Mn^{3+} \rightarrow Mn^{2+}$ redox reaction and deprotonation of WAT1 and Tyr34 for $H_2O_2$ formation during the $Mn^{2+} \rightarrow Mn^{3+}$ reaction[15,16]. These theoretical methods were the only way to study the simultaneous change of electron state and proton locations in the enzyme.

Neutron protein crystallography (NPC) is an emerging tool for analyzing hydrogen positions of biological macromolecules and possesses attributes that are useful in deciphering CPET mechanisms. In NPC, scattering of deuterium is on par with carbon, nitrogen, and oxygen, increasing the ability to locate proton positions for the entire enzyme. An additional advantage is that neutrons, unlike X-rays, do not alter the electronic state of metals[17]. The capability of proton visualization and amenability to enzyme redox control that NPC provides creates an avenue to investigate the CPET mechanism of MnSOD experimentally.

Here, we present room temperature neutron structures of human MnSOD at physiological pH in $Mn^{3+}$ and $Mn^{2+}$ states and reveal how the atomic locations of all protons in the enzyme active site change when the active site metal goes through a redox cycle. Direct experimental evidence is provided for deprotonation of a conserved glutamine residue and the feasibility of the proton transfer is explored with quantum calculations. Four other conserved residues are discerned to each have short-strong hydrogen bonds (SSHBs) and unexpected attributes that include an ionized tyrosine and a low-barrier hydrogen bond (LBHB) between a separate tyrosine and histidine across the dimer interface.

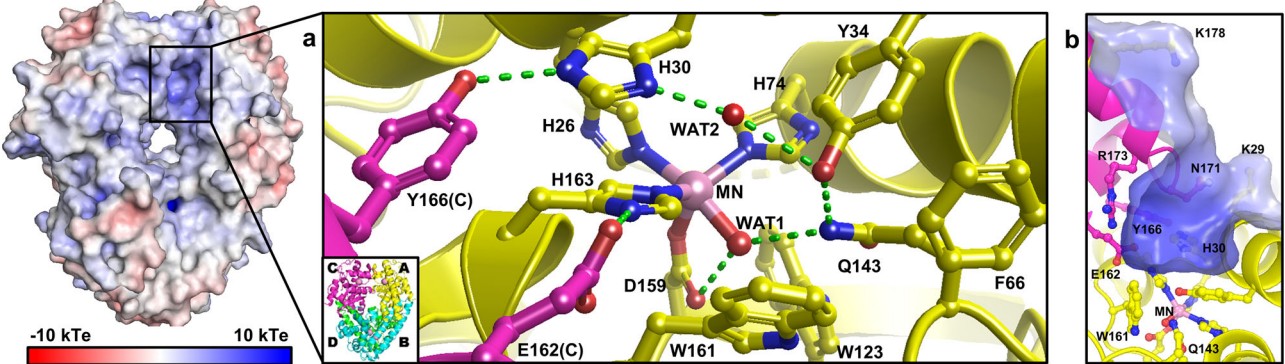

**Fig. 1 Structure of tetrameric human MnSOD. a** The active site of MnSOD is within a positively charged cavity formed from two adjacent subunits. The hydrogen bond network is denoted by green dashes and involves residues from both subunits. The inset indicates chain identity where A and B comprise the asymmetric unit while C and D are generated by symmetry. This chain coloring scheme is used in all the figures. Solvent and substrate accessibility is possible only through the ~5 Å gap between His30 and Tyr34. **b** Side-view of the active site cavity rotated approximately 90° relative to **a**. Structure is from PDB ID 5VF9 [10.2210/pdb5VF9/pdb][77].

A proposed CPET mechanism is reported from the culmination of data between $Mn^{3+}$SOD and $Mn^{2+}$SOD neutron structures.

## Results and discussion

**Direct evidence for CPETs with glutamine deprotonation.** To visualize the effect of the electronic state of the metal on the active site protons, all-atom, D-labeled neutron structures were obtained for $Mn^{3+}$SOD and $Mn^{2+}$SOD to resolutions of 2.20 and 2.30 Å, respectively. The two data sets collected were from the same crystal treated with appropriate oxidizing and reducing chemicals and the redox conditions were maintained during each data collection[18]. Neutron data at these resolutions are excellent and permitted ease in the visualization of deuterium atoms. The proton positions of the resting state, apo active sites were specifically sought to define the effects of metal redox state on the surrounding protons independent of substrate binding. We initially sought a crystallographic check for the success of the redox manipulations. WAT1, the Mn ligand, has historically been thought to obtain a proton ($^-OH \rightarrow HOH$) upon one-electron reduction of $Mn^{3+}$ to $Mn^{2+}$. This was inferred from crystallographic bond distances and QM calculations but otherwise had not yet been directly observed and confirmed[4,16,19–25]. Indeed, careful inspection of the neutron scattering length density between neutron data sets revealed changes in the protonation of WAT1 when the redox state of the Mn ion was altered (Fig. 2a, b). This was theoretically expected, verified our methods, and gave confidence in the data. To our knowledge, this is the first time the chemical reduction of an active site metal was visually observed to change the protonation state of a ligand.

For $Mn^{3+}$SOD, both chains of the asymmetric unit have a single nuclear $|F_o| - |F_c|$ density peak for the $D^1$ atom of WAT1 indicating the expected deuteroxide ($^-OD$) molecule and is supported by the Mn–O(WAT1) bond distance of 1.8 Å (Fig. 2a and Supplementary Fig. 1)[15,26]. The $^-OD$ acts as a hydrogen bond donor to the $O^{\varepsilon 2}$ of Asp159 with a distance of 1.9 Å whereas the O(WAT1) atom is acting as a hydrogen bond acceptor from $D^{\varepsilon 21}$ (Gln143) with a distance of 1.8 Å (Fig. 2a). The data for WAT1 of $Mn^{2+}$SOD instead have two nuclear $|F_o| - |F_c|$ density peaks for D atoms indicating the $^-OD$ was converted to $D_2O$ upon metal reduction, as expected. Further support suggesting a $D_2O$ molecule is seen with the Mn–O(WAT1) bond distance of 2.2 Å[15,26]. The $D^1$(WAT1) atom position is similar to that found in the $Mn^{3+}$SOD counterpart and hydrogen bonds with $O^{\varepsilon 2}$ of Asp159, albeit with a longer distance of 2.5 Å (Fig. 2b). Surprisingly, $D^2$ (WAT1) points toward Gln143 suggesting the WAT1 of $Mn^{2+}$SOD is acting as a hydrogen bond donor to Gln143. This means Gln143 is a hydrogen bond acceptor in $Mn^{2+}$SOD and its $D^{\varepsilon 21}$ atom is absent. Indeed, there is a lack of neutron scattering length density for $D^{\varepsilon 21}$ but not for $D^{\varepsilon 22}$. This interpretation was supported when attempts to model $D^{\varepsilon 21}$ led to negative $|F_o| - |F_c|$ neutron scattering length density. In the $Mn^{2+}$SOD structure, the hydrogen bond between $D^2$(WAT1) and $N^{\varepsilon 2}$(Gln143) is atypical with a bond distance of only 1.6 Å and O(WAT1)– $D^2$(WAT1)–$N^{\varepsilon 2}$(Gln143) angle close to 180°. These are characteristics of an SSHB, a type of hydrogen bond that is thought to stabilize particular enzymatic steps and enhance catalytic rates[27–29]. This is not to be confused with an LBHB, a type of SSHB where a proton is equidistant between heteroatoms[30]. SSHBs are noteworthy in several well-studied enzymes, such as α-chymotrypsin that utilizes an SSHB between the His and Asp of its catalytic triad[28]. This creates a ~7 kcal mol$^{-1}$ stronger interaction to substantially increase the kinetic rate. For $Mn^{2+}$SOD, the SSHB between WAT1 and Gln143 may contribute to the stability of the redox state and the high catalytic efficiency of the enzyme.

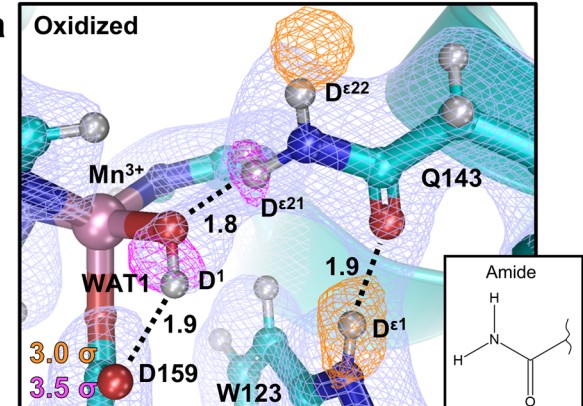

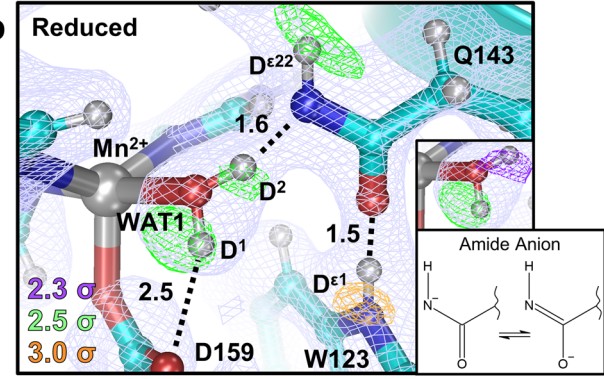

**Fig. 2 Proton transfer between Gln143 and the Mn-ligated solvent molecule WAT1. a** Neutron structure at the active site of $Mn^{3+}$SOD with magenta and orange omit $|F_o|-|F_c|$ difference neutron scattering length density displayed at 3.5σ and 3.0σ, respectively, and light blue $2|F_o| - |F_c|$ density (displayed at 1.0σ). Of note, the neutron scattering length of Mn is negative. Numbers are distances in Å. **b** Neutron structure at the active site of $Mn^{2+}$SOD with purple, green, and orange omit $|F_o| - |F_c|$ difference neutron scattering length density displayed at 2.3σ, 2.5σ, and 3.0σ, respectively, and light blue $2|F_o| - |F_c|$ density (displayed at 1.0σ). Both panels are for chain B. Only one chain is shown due to high structural similarities, see Supplementary Fig. 1 for chain A. Representations of the amide and amide anions are included.

The experimental data, therefore, suggest that Gln143 is undergoing deprotonation to form an amide anion and this is unusual because glutamine residues are not expected to act as weak acids since the p$K_a$ of primary amides are 16–18. However, p$K_a$ studies of less acidic secondary amides suggest p$K_a$ values may be depressed to 7–8 depending on how the amide group is polarized (i.e. charge delocalization)[31]. This is supported by the known event of proton exchanges occurring at the amide groups of protein backbones. Moreover, several enzyme studies suggest glutamine or asparagine-mediated proton transfers for catalysis and support the plausibility of a Gln143 → WAT1 proton transfer. For example, an asparagine residue has been suggested to be deprotonated in prenyltransferases due to significant polarization from close proximity to a metal cation[32]. Nakamura and coworkers[33] showed neutron data of cellulase Cel45A from *Phanerochaete chrysosporium* that revealed asparagine deprotonation that is instrumental for the proton relay of the enzyme. Infrared spectroscopy and computational calculations support the involvement of glutamine-mediated proton transfers in GTP hydrolysis by Ras-GAP and photoexcitation of photoreceptor proteins with the flavin-binding BLUF domain[34]. For MnSOD, the deprotonation of Gln143 for CPET to the active site ligand

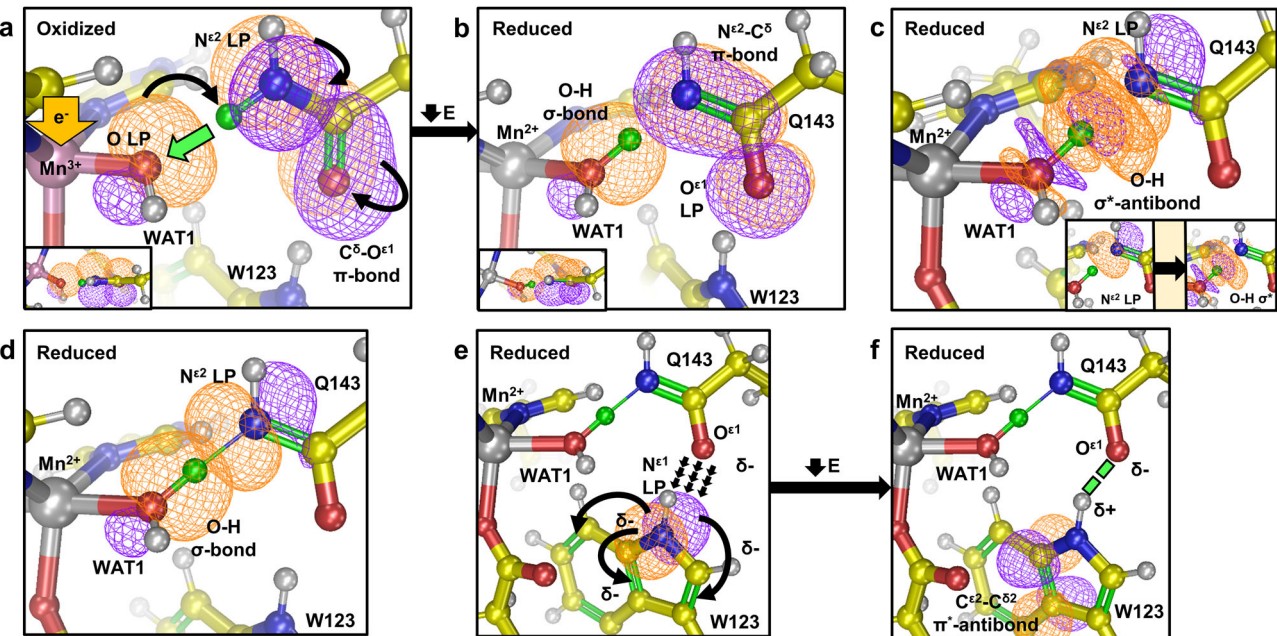

**Fig. 3 The suggested mechanism of Gln143 → WAT1 proton transfer utilizing bonding orbitals and optimal resonance forms calculated from chemist's localized property-optimized orbital (CLPO) analysis of the neutron structures.** Orange and purple contours indicate the positive and negative orbital wave functions, respectively. Curved arrows represent electron pushing. **a, b** The reduction of $Mn^{3+}$ to $Mn^{2+}$ instigates a strong covalent need for the exposed lone pair (denoted LP) of $^-OH$(WAT1). This is chemically remediated by the acquisition of a proton (green atom) from the proximal Gln143 amide. The dominant resonance structure of the amide anion is with a $N^{\varepsilon2}$–$C^\delta$ double bond as a result of $N^{\varepsilon2}$ LP delocalization from **a**. **c** Despite proton donation to WAT1, Gln143 still demonstrates covalent character with the proton (green atom). The new $N^{\varepsilon2}$ LP participates in electron density transfer to the $\sigma^*$-antibonding orbital of O–H(WAT1). Donor acceptor orbital analysis calculates a stabilizing energy of $1.4\ kcal\ mol^{-1}$ for this hyperconjugated interaction. The inset in the lower-right corner illustrates individual orbital representations. **d** Due to the hyperconjugation illustrated in **c**, the hydrogen bond between $N^{\varepsilon2}$(Gln143) and O–H(WAT1) has partial σ-bonding character contributing to and increasing the strength of the hydrogen bond characteristic of short-strong hydrogen bonds (SSHBs). CLPO calculations suggest the proton is covalently shared between WAT1 and Gln143, with percentages of 64 and 36%, respectively. **e, f** The increased electronegative character of $O^{\varepsilon1}$ electrostatically polarizes Trp123. This is achieved through the stabilizing delocalization of the $N^{\varepsilon1}$(Trp123) LP into the conjugated ring. The major stabilizing interaction decreases energy by $13.52\ kcal\ mol^{-1}$ and is the donation of electron density from the $N^{\varepsilon1}$(Trp123) LP into the adjacent $C^{\varepsilon2}$–$C^{\delta2}$ $\pi^*$-antibonding orbital. The polarization of Trp123 allows an SSHB between Gln143 and Trp denoted by the green dashes. The hyperconjugation and polarization of Trp123 are thought to contribute to the stability of the amide anion.

has not been observed before although it does explain the high efficiency of the enzyme as a result of this internal proton source.

Density functional theory (DFT) calculations of the active site using the neutron structures support our interpretation of the neutron scattering length density for deprotonation of Gln143. Calculations used the atoms of the residues shown in Fig. 1. Chemist's localized property-optimized orbital (CLPO) analysis (akin to Natural Bond Orbitals[35–37]) was used to evaluate the interactions of Gln143 and WAT1 (ref. [38]). Reduction of $Mn^{3+}$ to $Mn^{2+}$ increases the electronegative character of the O(WAT1) lone pair facing the proximal amide proton of Gln143 (green atom, Fig. 3a). This polarization of $^-OH$(WAT1) increases its basicity and chemically allows the abstraction of the amide proton from glutamine. In terms of hard and soft Lewis acid and base (HSAB) theory, the transition of the metal acid and solvent base from ($Mn^{3+}$–$^-OH$) to ($Mn^{2+}$–$OH_2$) corresponds to both the acid and base becoming softer. The bonds of the deprotonated amide rearrange to stabilize its new negatively charged state (Fig. 3b). The $O^{\varepsilon1}$ atom of Gln143 bears the most electronegative charge, and calculations suggest less covalent electrons for the $O^{\varepsilon1}$–$C^{\varepsilon1}$ bond compared to that of $N^{\varepsilon2}$–$C^{\varepsilon1}$, with bond orders of 1.33 and 1.52, respectively (Supplementary Table 1). The presence of an SSHB is supported as well since the deprotonated $N^{\varepsilon2}$ still has covalent character with the donated proton (Fig. 3c). Donor–acceptor orbital analysis indicates electron density is transferred from the $N^{\varepsilon2}$ lone pair orbital to the $\sigma^*$-antibonding orbital of O–H(WAT1) and is a $1.4\ kcal\ mol^{-1}$ stabilizing

hyperconjugation interaction (Supplementary Table 2). As a result, the interaction between $N^{\varepsilon2}$ and the donated proton demonstrates partial σ-bonding character. The extent of covalence is 36% for the $N^{\varepsilon2}$–H bond and 64% for the O–H bond (Supplementary Table 3). Using the same QM methods, water molecules participating in normal hydrogen bonding have 12% covalence between the donor H and acceptor O atoms. Altogether, quantum calculations support both Gln143 deprotonation and the presence of an unusual hydrogen bond between WAT1 and Gln143.

A previously undetected change in the interaction of Gln143 with the neighboring Trp123 was observed. An SSHB of 1.5 Å is seen between the $O^{\varepsilon1}$ of Gln143 and $D^{\varepsilon1}$ of Trp123 in the reduced state (Fig. 2b). The same hydrogen bond is seen in $Mn^{3+}$SOD at a distance of 1.9 Å (Fig. 2a). The neutron data, therefore, support the notion that $O^{\varepsilon1}$(Gln143) harbors a stronger electronegative character during the $Mn^{2+}$ redox state. This may be a consequence of Gln143 deprotonation to the amide anion during the $Mn^{3+}$ → $Mn^{2+}$ redox reaction and negative charge stabilization of its $O^{\varepsilon1}$ atom through hydrogen bonding with Trp123. Charge stabilization is likely to be important for glutamine deprotonation as amide groups are known to deprotonate at neutral pH when electronegatively polarized at the carbonyl O atom[31]. Trp123 is especially competent at charge stabilization due to its own ability to polarize. CLPOs suggest that lone pair electrons of $N^{\varepsilon1}$(Trp123) delocalize into the highly conjugated aromatic ring of Trp123 when the glutamine amide is

deprotonated and is stabilizing (Fig. 3e). Donor–acceptor orbital analysis calculates that the major stabilizing interaction is the donation of $N^{\varepsilon 1}$(Trp123) lone pair electron density into the $C^{\varepsilon 2}$–$C^{\delta 2}$ $\pi^*$-antibonding orbital (Fig. 3f) and decreases energy by 13.5 kcal mol$^{-1}$ (Supplementary Table 2). This also permits the SSHB between $O^{\varepsilon 1}$(Gln143) and $H^{\varepsilon 1}$(Trp123). Quantum calculations indicate an important role for Trp123 in the deprotonation of Gln143.

If an O(WAT1)–$D^{\varepsilon 21}$(Gln143)–$N^{\varepsilon 2}$(Gln143) interaction is needed for redox cycling of Mn, mutation of Gln143 or a nearby residue that may stabilize amide deprotonation should affect catalysis. In the literature, the Gln143Asn mutant has nearly ablated catalysis in both redox states while Trp123Phe can perform catalysis for $Mn^{3+} \rightarrow Mn^{2+}$ at deficient lower rates (20–50%) but not at all for the $Mn^{2+} \rightarrow Mn^{3+}$ transition[14,39]. The effect of these mutations suggests that Gln143 is central to catalytic activity while Trp123 is most significant for the $Mn^{2+} \rightarrow Mn^{3+}$ half of the redox cycle. The detrimental effects for the $Mn^{2+}$ state due to mutating residue Trp123 may therefore reflect their role in stabilizing the Gln143 amide anion. Indeed, the kinetic behaviors of these mutants were puzzling in past

studies but amide proton transfer potentially explains them[12,39–41]. Glutamine at the position of Gln143 is conserved in all isoforms of MnSODs and prokaryotic FeSODs. A closer WAT1-Gln distance correlates with increased redox potentials and catalytic rates[42]. This is perhaps because of an enhanced ability for proton transfers between O(WAT1) and $N^{\varepsilon 2}$(Gln143). Past mutagenesis studies, differences in catalytic rates among isoforms, and the high catalytic rate of MnSOD may be explained by Gln143 serving as an internal proton source for CPET via amide deprotonation.

**Tyr34 has an unusual p$K$a and SSHBs with the Gln143 anion.** Tyr34 is positioned near the active site solvent channel, hydrogen-bonded to Gln143 (Fig. 1a), and has been hypothesized to be a proton source for MnSOD CPET[12]. For $Mn^{3+}$SOD, Tyr34 does not have a nuclear peak for its hydroxyl proton (Fig. 4a and Supplementary Fig. 3). Interestingly, for one of the active sites, deprotonated Tyr34 is making a very strong hydrogen bond with a nearby solvent molecule (designated WAT2) with a 2.3 Å distance between heteroatoms $O^{\eta}$(Tyr34) and O(WAT2) (Fig. 4a). While the

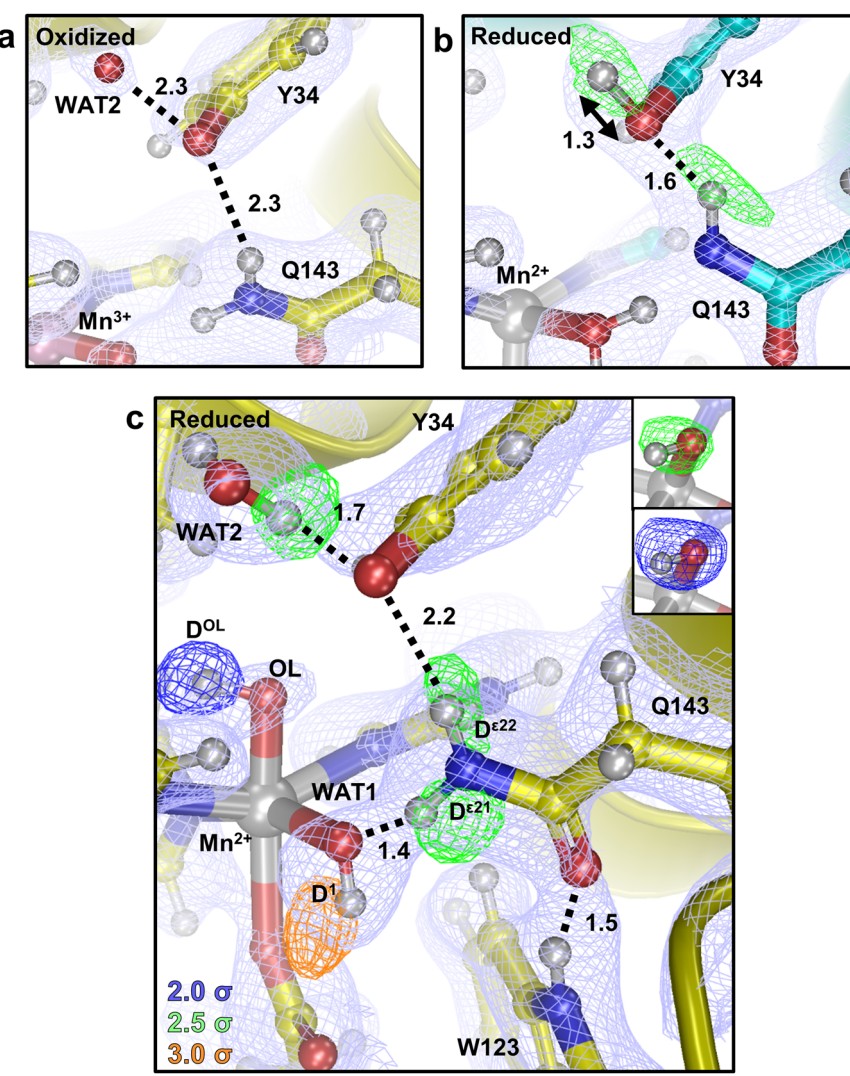

**Fig. 4 Protonation and active site coordination of MnSOD. a, b** Neutron structures of five-coordinate $Mn^{3+}$SOD and $Mn^{2+}$SOD. Green omit $|F_o|-|F_c|$ difference neutron scattering length density is at 3.0$\sigma$. **c** The six-coordinate $Mn^{2+}$SOD active site of Chain A with green omit $|F_o|-|F_c|$ difference density displayed at 2.0$\sigma$, 2.5$\sigma$, and 3.0$\sigma$ for blue, green, and orange contours, respectively. $|F_o|-|F_c|$ difference density is for individual D atoms except for the upper-right corner that is for both atoms of the anionic oxygen ligand (OL). Light blue 2$|F_o|-|F_c|$ neutron scattering length density is displayed at 1.0$\sigma$. Numbers are distances in Å. For the oxidized state, only one chain is shown due to high structural similarities, see Supplementary Fig. 2 for chain B.

deuteriums of WAT2 could not be discerned, the distance is characteristic of an SSHB where Tyr34 may be poised to accept a proton. This interpretation is supported by CLPO analysis from DFT, with 80/20 covalent sharing of the proton (Supplementary Table 3). For $Mn^{2+}$SOD, a nuclear peak for the hydroxyl proton is present but not where it is expected. Refinement with the ideal 0.97 Å $O^{\eta}$–$D^{\eta}$ distance for Tyr34 persistently demonstrates residual $|F_o|$ −$|F_c|$ difference density (Supplementary Fig. 3) uncharacteristic of the other MnSOD tyrosine residues. Between refinement cycles, restraints for this Tyr34 were incrementally loosened from the 0.97 Å ideal hydroxyl distance until the $|F_o|$−$|F_c|$ difference density was appropriately absent and the $B$ factors were comparable to other nearby atoms. This yielded an unusual $O^{\eta}$–$D^{\eta}$ bond length of 1.3 Å that points towards the solvent channel (Fig. 4b). Intriguingly, the $O^{\eta}$(Tyr34) atom participates in a strong 1.6 Å hydrogen bond with $D^{\varepsilon 22}$(Gln143) in the $Mn^{2+}$SOD structure and is significantly different than the corresponding hydrogen bond distance of 2.3 Å in $Mn^{3+}$SOD (Fig. 4a, b). This SSHB may potentially be explained by the increased polarization of Gln143 in $Mn^{2+}$SOD from deprotonation to the amide anion leading to a stronger hydrogen bond interaction with Tyr34. It is unclear whether this interaction contributes to the observed changes in Tyr34 protonation though it may stabilize the amide anion of Gln143. The experimental data, therefore, show that Tyr34 is capable of gaining and losing a proton at physiological pH indicative of an unusual $pK_a$, and participates in atypical hydrogen bonding.

Our experimental data for Tyr34 potentially shines a light on the unexplained observations of previous studies investigating its role in catalysis[8,12,39]. Tyr34 has been speculated to be the proton donor to WAT1 for CPET during the $Mn^{3+} \rightarrow Mn^{2+}$ reaction though this conflicts with the pH independence of the reaction between values of 6 and 10 (refs. [8,43]). This was puzzling because CPET mechanisms are expected to have pH dependence as a result of the proton transfer part of their catalysis and Tyr34 is the closest titratable residue. Instead, the MnSOD neutron data suggest that the proton donor to WAT1 is internally sourced from Gln143 without the direct involvement of solvent and cannot be Tyr34 due to its observed deprotonation in the $Mn^{3+}$ state.

An ionized tyrosine residue at physiological pH is unusual though is potentially explained by polarization from the pronounced positive charge of the metal. Such an effect has been visualized in studies of human carbonic anhydrase II (HCA II), a metalloenzyme with diffusion-limited catalytic efficiencies like MnSOD. For HCA II, joint neutron crystallography and NMR demonstrate a tyrosine residue with a $pK_a$ of 7.10 ± 0.10 at the active site[44]. The catalytic role of an ionizable Tyr34 for MnSOD is prominent during the $Mn^{2+} \rightarrow Mn^{3+}$ redox cycle and is supported by the inability of the Tyr34Phe mutant to catalyze this step of the reaction[45]. Since Tyr34 gains a proton during the $Mn^{3+} \rightarrow Mn^{2+}$ redox cycle and loses a proton during the $Mn^{2+} \rightarrow Mn^{3+}$ cycle, it is conceivable that Tyr34 serves as the source for one of the two protons involved in the protonation of the substrate to $H_2O_2$. Indeed, spectroscopic data of the Tyr34Phe mutant suggest a prolonged binding of a species to the metal but could also be a result of the inability of Gln143 to deprotonate without stabilization from the Tyr34 hydroxyl group[12]. Our crystallographic neutron data have shed light on the perplexing role of the conserved Tyr34 residue.

**Serendipitous ligand binding to $Mn^{2+}$SOD helps explain catalysis.** Previous spectroscopic studies have suggested the sixth-coordinate binding of a ligand ($^{-}OH$, $^{-}F$, or $^{-}N_3$) to $Mn^{2+}$SOD coinciding with lengthening or displacement of the $Mn–O^{\delta 2}$ (Asp159) bond essentially converting it to a 5-coordinate active

site spectrophotometrically[19,46]. The chemical purpose of this modification in coordination has been unclear. A sixth-coordinate $^{-}OH$-bound complex has been difficult to visually confirm due to poor scattering of hydrogen by X-rays[47]. Serendipitously, one of the active sites of the $Mn^{2+}$SOD neutron structure has density for an unexpected sixth-coordinate $^{-}OD$ ligand (designated OL for anionic oxygen ligand, Fig. 4c), and the $Mn–O^{\delta 2}$(Asp159) bond is stretched from 1.95 to 2.44 Å. Of note, the buffer system did not include $^{-}OD$, and only appropriate ratios of $K_2DPO_4$ and $KD_2PO_4$ were used to achieve a pD of 7.8. Yet the observed nuclear scattering length density for deuterium and the 1.84 Å Mn–O distance of OL supports the identification of OL as $^{-}OD$[20]. OL is well ordered with a $B$-factor that is comparable to other atoms of the active site (Supplementary Table 4). Further validation was sought with DFT geometry optimizations of the observed neutron structure, $Mn^{2+}$SOD with sixth-coordinate $^{-}OH(OL)$, and suggests stability of the six-coordinate complex while replacing OL with $H_2O$ causes disassociation into five-coordinate $Mn^{2+}$SOD. The observed differences between the $Mn^{2+}$SOD active site (Fig. 4b, c) may be a consequence of the crystallographic asymmetric subunits having different capacities of solvent accessibility (Supplementary Fig. 4). Nevertheless, this active site supports past hypotheses of $Mn^{2+}$-binding $^{-}OH$ at the sixth-coordinate position[19,47] and occurs in tandem with the lengthening of the $Mn–O^{\delta 2}$(Asp159) as proposed by the associative displacement ligand binding model[46].

Maliekal and coworkers[19] previously hypothesized that binding of a $^{-}OH$ ligand could be the result of an electronegative deprotonated Tyr34 and a electropositive Mn ion polarizing a water molecule to promote deprotonation. Indeed, Tyr34 is observed to have no density for a hydroxyl proton (Fig. 4c). Since Tyr34 is deprotonated, the proton acceptor of the water molecule is likely to be another nearby residue, perhaps His30. The subsequent OL coordination to $Mn^{2+}$ could be involved in Gln143 deprotonation. The six-coordinate active site has Gln143 in the canonical amide form making a $D^{\varepsilon 21}$(Gln143)-O(WAT1) 1.4 Å SSHB (Fig. 4c). This distance is even shorter than the N (Gln143)-$D^2$(WAT1) SSHB in apo $Mn^{2+}$SOD (Fig. 2b) and suggests a very strong hydrogen bonding interaction that is likely to be partially covalent (Supplementary Table 5). Trp123 forms a stabilizing 1.5 Å SSHB with $O^{\varepsilon 1}$(Gln143) as observed at the other subunit (Fig. 2b) and is thought to support deprotonation of Gln143 (Fig. 3e, f). These structural characteristics indicate that binding of OL probably contributes to the future deprotonation of Gln143. Here, binding by an anionic OL would lower the positive charge of $Mn^{2+}$ and electronegatively polarize $^{-}OD$ (WAT1) to initiate covalent bonding with the proximal $D^{\varepsilon 21}$ (Gln143) proton.

**His30 and Tyr166 form a LBHB indicating unusual $pK_a$s.** His30 is positioned at the active site channel (Fig. 1) and also shows interesting changes in protonation between redox states. The two nitrogen atoms of His30 that are potential sites for protonation changes are strategically positioned. The $N^{\delta 1}$ atom is solvent accessible and is hydrogen bonding to WAT2 while the $N^{\varepsilon 2}$ atom is not accessible to solvent and hydrogen bonds with the buried Tyr166 from the adjacent subunit (Fig. 1a). For $Mn^{2+}$SOD, both chains show strong omit $|F_o|$−$|F_c|$ difference density for protonation of $N^{\delta 1}$ (Fig. 5a, b). Interestingly, the D atom between Tyr166 and His30 refines to a position that is nearly equidistant between $O^{\eta}$ and $N^{\varepsilon 2}$ and has elongated omit $|F_o|$−$|F_c|$ difference density that is unusual for a typical hydrogen bond. This is characteristic of a LBHB, a type of SSHB where a proton is transiently shared between heteroatoms[48]. LBHBs are thought to be instrumental in catalysis by stabilizing enzyme states by as

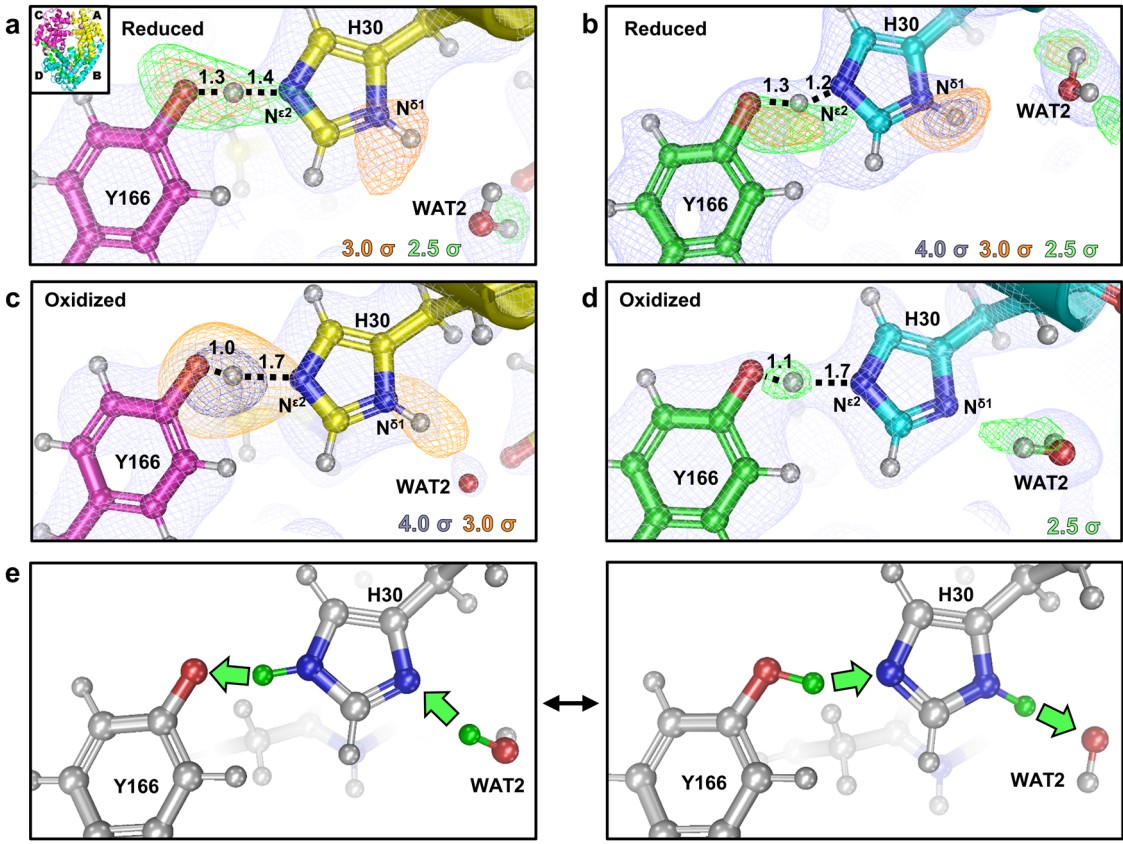

**Fig. 5 Protonation states observed for His30 and Tyr166 among the different chains of Mn$^{2+}$SOD and Mn$^{3+}$SOD. a**, **b** Neutron structures at the His30-Tyr166 interface of Mn$^{2+}$SOD. **c**, **d** Neutron structures at the His30-Tyr166 interface of Mn$^{3+}$SOD. The inset of **a** indicates chain identity. Light blue $2|F_o|-|F_c|$ neutron scattering length density is displayed at 1.0σ. Purple-blue, orange, and green omit $|F_o|-|F_c|$ difference densities are displayed at contours of 4.0σ, 3.0σ, and 2.5σ, respectively. **e** Overall interpretation for a proton shuttle involving Tyr166 and His30. Numbers are distances in Å.

much as 20 kcal mol$^{-1}$ leading to a crucial acceleration of kinetic rates[30,49]. Indeed, Tyr166Phe and His30Gln single mutants both decrease $k_{cat}$ by ~30-fold and $k_{cat}/K_m$ by ~10-fold[13,50]. Another defining feature of LBHBs is the p$K_a$ difference between heteroatoms is near zero[51]. Buried amino acid residues can have elevated or depressed p$K_a$s[52,53]. His30 and Tyr166 may be following these qualities. The p$K_a$ of His30 would be elevated from its solvated value of ~6, and the p$K_a$ of Tyr166 would be depressed from its solvated value of ~10. Since N$^{ε2}$(His30) is solvent inaccessible along with Tyr166, the LBHB may be supported by residing within an enclosed environment.

For Mn$^{3+}$SOD, the D atom between His30 and Tyr166 refines with the average position on Tyr166, and its hydrogen bond distance with N$^{ε2}$(His30) is of typical length (Fig. 5c, d). Despite the refinement position of the D atom at Chain A, the omit $|F_o|-|F_c|$ difference density is pronounced and elongated towards N$^{ε2}$(His30) and may reflect the movement of the proton (Fig. 5c). Similarly, the N$^{δ1}$-bound proton of Chain A has strong and elongated omit $|F_o|-|F_c|$ difference density towards WAT2 that can be interpreted as proton exchange with solvent. For Chain B, there is no difference density for a proton on N$^{δ1}$(His30), a stark contrast with Chain A (Fig. 5d) as well as both chains of Mn$^{2+}$SOD (Fig. 5a, b). Chain B of Mn$^{3+}$SOD is also the only one with WAT2 positioned to act as hydrogen bond donor to N$^{δ1}$(His30) rather than a hydrogen bond acceptor, which would agree with the notion that N$^{δ1}$(His30) is deprotonated. With N$^{δ1}$ and N$^{ε2}$ of His30 both deprotonated, the imidazole group puzzlingly appears absent of protons which is chemically infeasible. Differences in $B$ values between the subunits of solvent-exposed residues could degrade the quality of the density but this does not explain the absence of proton density at N$^{δ1}$(His30) for

Chain B. In fact, these residues in Chain B have lower refined $B$ values than in Chain A (Supplementary Table 4). By comparing the density of the D atom between His30 and Tyr166 with that of the other chain (Fig. 5c), it is conceivable that the D atom appearing to belong to Tyr166 has movement to N$^{ε2}$(His30) and this is the most plausible explanation. The chemical function may be to modulate protonation at the solvent accessible N$^{δ1}$ of His30. Protonation of N$^{ε2}$(His30) would coincide with deprotonation of N$^{δ1}$(His30) leading to shifting between singly N$^{δ1}$- or N$^{ε2}$-protonated tautomers (Fig. 5e).

Similarly, unusual hydrogen bonding between residues has been observed in other enzymes. The most notable are histidines found in catalytic triads of proteases, where changes in protonation of one nitrogen could be tied to the protonation state of the other with the help of an SSHB interaction[54]. Investigating the literature for other enzymes utilizing a tyrosine–histidine pair in catalysis reveals the metalloenzyme Photosystem II (PSII) that utilizes CPETs[55]. The tyrosine–histidine pair of (PSII) appears to have an SSHB that needs to be maintained for catalysis with measured p$K_a$ values ranging between 7.3 and 8.0. For MnSOD, the observed atypical hydrogen bonding observed across the dimer interface between Tyr166 and His30 is probably important in the enzymatic mechanism.

Previous mutagenesis investigations suggest that the Tyr166 and His30 interaction is needed for catalysis and support the interpretation of proton transfers occurring between N$^{ε2}$(His30) and O$^{η}$(Tyr166) that may coincide with changes in protonation at N$^{δ1}$(His30). His30Gln is the only His30 mutant that has been studied that maintains the hydrogen bonding at the active site and does not significantly affect the

positions of other residues at the active site[13,50]. Kinetically, the His30Gln rate for $k_1$ (Mn$^{3+}$ → Mn$^{2+}$) is 38% of the wild type while $k_2$ (Mn$^{2+}$ → Mn$^{3+}$) is 72%[56]. The rates indicate an important role for His30 $k_1$ that may correspond with our observations of its changes in protonation in Mn$^{3+}$SOD. It should be noted that previous studies refrain from attributing differential protonations to His30 due to the similar redox potentials between wild type (393 ± 29 mV) and His30Gln (380 ± 30 mV)[13,50,57]. However, the investigations do not consider whether compensatory protonations or deprotonations occur at nearby residues as a result of the mutation giving the appearance of an inconsequential effect. Indeed, the Tyr34Phe mutant also has an insignificant change of redox potential (435 ± 30 mV), but has a nearly identical rate for $k_1$ (37% of wild type) as His30Gln[50,56]. Drawing an inference from the Tyr166Phe mutant is difficult because hydrogen bonding and side-chain conformations are significantly changed at the active site, but this mutant has nearly identical measurements of redox potential (436 ± 10 mV) compared to Tyr34Phe, indicating a synonymous effect to the charge of the active site[50,56]. The neutron data for His30 and Tyr166 help explain past observations of MnSOD mutants that were perplexing and ties together changes of protonation state with kinetic and redox potential measurements.

The neutron structures of Mn$^{3+}$SOD and Mn$^{2+}$SOD have revealed active site states that have unique configurations of protonations and hydrogen bonds. Two of the configurations are the enzymatic resting states for Mn$^{3+}$ and Mn$^{2+}$ characterized by a five-coordinate Mn[46]. For the oxidized resting state that is described by both chains of the Mn$^{3+}$SOD neutron structure (Fig. 6a), the proton bridging His30 and Tyr166 appears to be moving and this suggests the possibility of Tyr166 alternating between an ionized or protonated form and a deprotonated or protonated N$^{\varepsilon 2}$(His30). N$^{\delta 1}$(His30) is also observed to be both deprotonated or protonated (Fig. 5c, d). For simplicity, only one protonation form of His30 and Tyr166 is shown (Fig. 6a). An SSHB is seen between WAT2 and an ionized Tyr34 while Gln143 is in the canonical amide form hydrogen bonding to WAT1 as a −OH molecule. For the reduced resting state described by chain B of Mn$^{2+}$SOD (Fig. 6b), Tyr166 and His30 now share the proton originating from N$^{\varepsilon 2}$(His30) with a LBHB, His30 is protonated on N$^{\delta 1}$, and Tyr34 is no longer ionized. Gln143 and WAT1 have undergone a proton transfer where WAT1 is now H$_2$O and Gln143 is an amide anion. Gln143 forms SSHBs with WAT1 and Trp123 presumably to stabilize its ionized form. The third active site state is a six-coordinate Mn$^{2+}$ with −OH-bound opposite

Asp159 (Fig. 6c) described by chain A of Mn$^{2+}$SOD. The protonations and SSHBs observed indicate an active site that is catalytically present before the five-coordinate reduced resting state but after a Mn$^{3+}$ → Mn$^{2+}$ redox transition. Tyr166 and His30 share a LBHB and His30 is N$^{\delta 1}$ protonated, like the reduced resting state, while Tyr34 is ionized like the oxidized resting state. Gln143 in the amide form but forms an SSHB with −OH(WAT1) and also Trp123. In total, five sites undergo changes in protonation state, Tyr166, His30, Tyr34, Gln143, and WAT1 that occur with four instances of SSHBs between pairs Tyr166-His30 (a LBHB), WAT2-Tyr34, Gln143-WAT1, and Gln143-Trp123.

In total, the present work provides details for some of the critical aspects of the CPET mechanism of human MnSOD. Through neutron diffraction, direct evidence is observed for (1) an internal protonation mechanism via glutamine deprotonation to the Mn-bound solvent molecule (WAT1) supported by quantum calculations; (2) an SSHB of Trp123 with the anionic form of Gln143 that stabilizes the anion; (3) changes in protonation of Tyr34 that interacts intimately with a solvent molecule (WAT2) when ionized, and (4) alternate protonation states and a LBHB for His30 and Tyr166 across the dimer interface. It is evident that the residues of the active site have unusual p$K_a$s that are likely a consequence of a polarized environment provided by the metal and limited solvent accessibility. As a result of obtaining neutron structures for both Mn$^{3+}$ and Mn$^{2+}$ states, we built a suggested mechanism that details the changes of protonation states as a result of the Mn ion gaining or losing an electron. For the mechanism, the superoxide substrate is not present but is represented strictly as its donation or abstraction of electrons due to the lack of experimental evidence for its precise binding site.

Starting from the five-coordinate Mn$^{3+}$ resting state (Fig. 7a), Mn$^{3+}$ acquires an electron (in reality from the substrate superoxide) that coincides with N$^{\delta 1}$(His30) acquiring a proton from the nearest solvent molecule (the crystallographic position of WAT2) and Tyr166 gaining partial covalent character from the proton of N$^{\varepsilon 2}$(His30) to form a LBHB (Fig. 7b). The Mn$^{2+}$ active site then binds −OH(OL) to form a six-coordinate Mn$^{2+}$ complex and may be the same solvent molecule that donated a proton to His30 (Fig. 7c). The depression of Mn$^{2+}$ positive charge through OL binding causes negative polarization at WAT1 and triggers proton abstraction from Gln143. Consequently, the

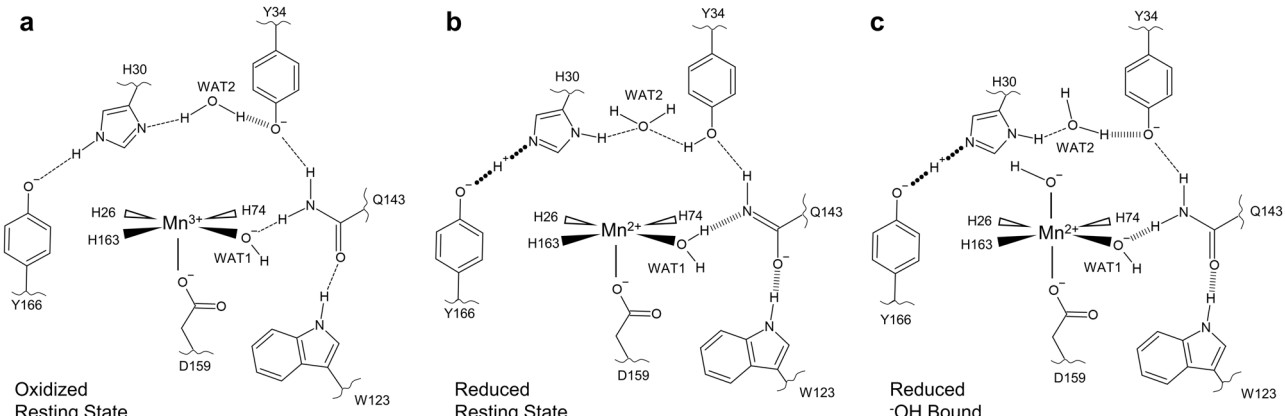

**Fig. 6 Summary of the unique active site configurations observed in neutron structures of Mn$^{3+}$SOD and Mn$^{2+}$SOD. a** Five-coordinate oxidized resting state. Of note, the data between the two Mn$^{3+}$SOD chains suggest the proton bridging His30 and Tyr166 may be moving. Additionally, the His30 N$^{\delta 1}$ atom closest to WAT2 is observed to be both deprotonated and protonated. For brevity, the most plausible configuration is shown. **b** Five-coordinate reduced resting state. **c** Reduced six-coordinate state with −OH-bound opposite Asp159. Dashed lines represent normal hydrogen bonds, wide dashed lines are short-strong hydrogen bonds, and round-dotted lines are low-barrier hydrogen bonds. Portrayal of hydrogen bond lengths in 2D are not representative of those seen experimentally in 3D.

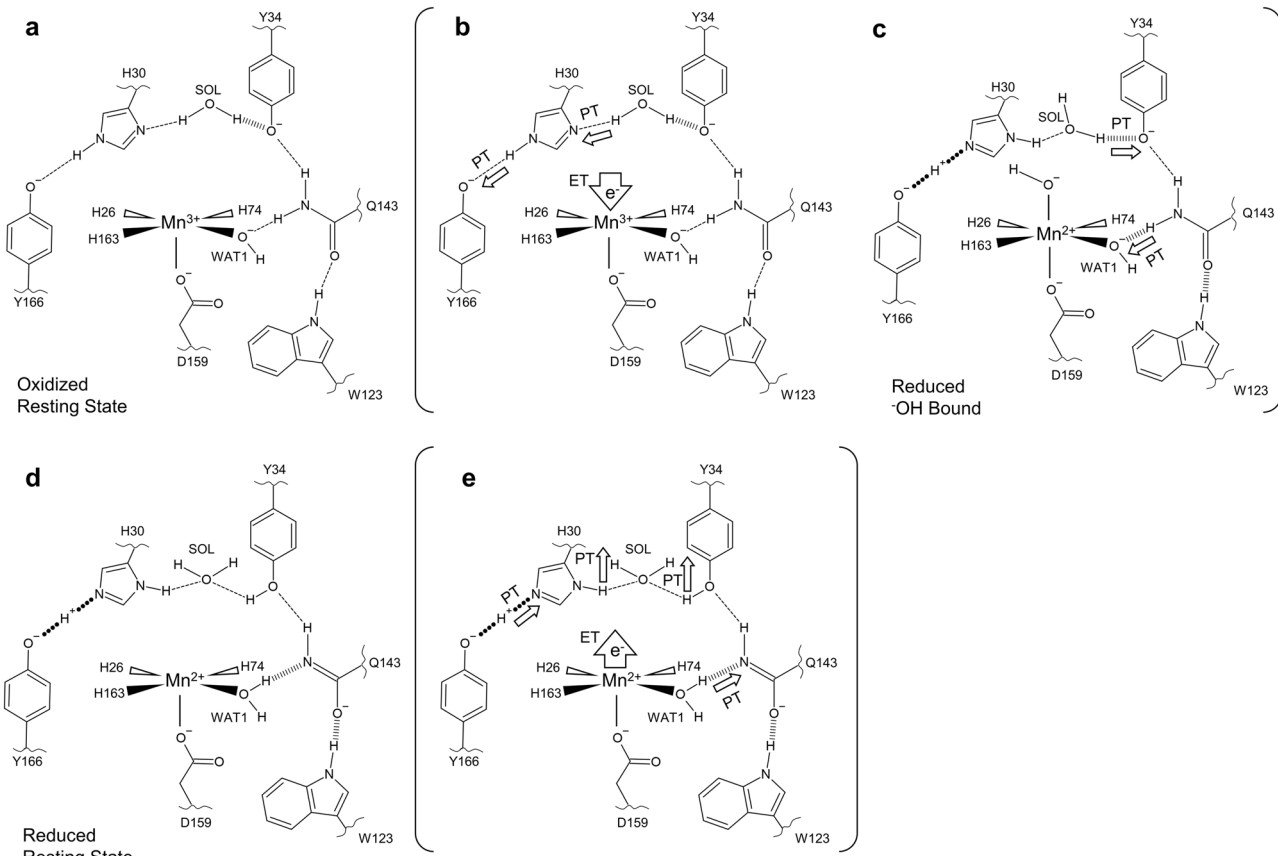

**Fig. 7 A suggested mechanism for MnSOD-active site proton transfers that coincide with electron gain or loss at the Mn ion.** Solvent, substrate enters the active site through the His30 and Tyr34 gateway. SOL represents the closest free solvent molecule typically found at the crystallographic site of WAT2 and is replenished from bulk solvent upon enzymatic use. **a** The five-coordinate oxidized resting state. **b** Concerted proton-electron transfer (CPET) is initiated by acquisition of an electron by $Mn^{3+}$ that is coupled to both $N^{\delta1}$(His30) proton gain from solvent and $N^{\varepsilon2}$(His30) proton sharing to buried Tyr166. **c** Driven by the new electrostatic environment, a $^-$OH molecule binds sixth-coordinate to $Mn^{2+}$. This suppresses the positive charge of $Mn^{2+}$ and polarizes WAT1 to become more negative and instigates proton gain from Gln143 deprotonation. The protonation at WAT1 causes electronegative polarity to instead be localized to the bound $^-$OH ligand, denoted OL. OL is subsequently protonated and departs from the active site while Tyr34 becomes protonated to form the five-coordinate reduced resting state. **d** The five-coordinate reduced resting state. **e** Electron loss by substrate coincides with the loss of protons at His30 and Tyr34 that are presumed to be acquired by the substrate. Tyr166 and WAT1 donate protons to His30 and Gln143, respectively, as a consequence of the net charge changes to regenerate the state in **a**. Bracketed panels indicate intermediary catalytic steps between the two resting states. The portrayal of hydrogen bond lengths in 2D are not representative of those seen experimentally in 3D.

WAT1–Gln143 interaction is more stabilizing and the electronegative polarity now localizes to OL and departs by its protonation by either solvent or Tyr34 as suggested by the Miller group[19]. Regardless, Tyr34 becomes non-ionized before the formation of the five-coordinate, reduced resting state (Fig. 7d). When the substrate is present, the steps of Fig. 7a–c describe the first CPET where proton and electron transfers are coupled to an extent where they cannot be differentiated with kinetic measurements[43]. The second CPET involves substrate obtaining an electron from $Mn^{2+}$ and protons from His30 and Tyr34 to form the $H_2O_2$ product (Fig. 7e). The changes of charge due to proton and electron departure from the active site causes Gln143 to accept the same proton it previously donated to WAT1 and $N^{\varepsilon2}$(His30) ceasing its proton sharing with Tyr166 to regenerate five-coordinate $Mn^{3+}$.

Altogether, the suggested mechanism utilizes two internal proton transfers where the protons move back-and-forth within the active site and two external proton transfers where the protons originate from solvent molecules to ultimately be consumed to form the product. The proton transfer between WAT1 and Gln143 is central to the mechanism as it permits the cyclic nature of catalysis. From this study, we have revealed, to our knowledge, the first direct coupling of electronic states to protonation states for an oxidoreductase. It is evident that the CPET mechanism of MnSOD is not straightforward as is exemplified by the previous elusiveness of the proton source for WAT1. Tyr34 was assumed to be the donor, but our data instead indicate an unusual and unexpected proton transfer from the Gln143 amide. Likewise, Trp123, His30, and Tyr166 were not assumed to be involved in catalysis. As this is just one biologically relevant oxidoreductase in a sea of many, finding the protonation states at the active sites of other oxidoreductases may reveal further unusual mechanisms for CPET.

## Methods

**Perdeuterated expression, purification, and crystallization.** For deuterated protein expression of MnSOD, the pCOLADuet-1 expression vector harboring full-length cDNA of *MnSOD* was transformed into *Escherichia coli* BL21(DE3) cells. Transformed cells were grown in $D_2O$ minimal media within a bioreactor vessel using $D_8$-glycerol as the carbon source[58]. Induction was performed with 1 mM isopropyl β-D-thiogalactopyranoside, 8 mM $MnCl_2$, and fed $D_8$-glycerol until an $OD_{600}$ of 15.0. Expression was performed at 37 °C for optimal Mn ion metal incorporation[59]. Harvested cell pastes were stored at −80 °C until purification. For

protein purification, cells were resuspended in a solution of 5 mM $MnCl_2$ and 5 mM 3-(*N*-morpholino)propanesulfonic acid (MOPS), pH 7.8. Clarified lysate was incubated at 55 °C to precipitate contaminant proteins that were subsequently removed by centrifugation. Next, soluble protein was diluted with an equal volume of 50 mM 2-(*N*-morpholino)ethanesulfonic acid (MES) pH 5.5 yielding a final concentration of 25 mM. Measurement of pH verified a value of 5.5 after dilution. Protein was applied onto a carboxymethyl sepharose fast flow column (GE Healthcare) and eluted with a 0-500 mM sodium chloride gradient that contained 50 mM MES pH 6.5 before concentration to 23 mg mL$^{-1}$. Crystallization was performed using a nine-well glass plate and sandwich box setup (Hampton Research) and the reservoir solution consisted of 1.9 M potassium phosphate adjusted to pH 7.8 by varying ratios of $KH_2PO_4$ and $K_2HPO_4$. The crystallization drop was a mixture of 60 μL concentrated protein solution (within a buffer of 50 mM MES pH 6.5) and 40 μL of the reservoir solution. Crystals grew up to 0.5 mm$^3$ after 6 weeks at 23 °C. Purification and crystallization were performed with hydrogenated reagents.

**Redox manipulation of perdeuterated MnSOD crystals**. Before redox manipulation, initial deuterium exchange of crystals was performed by vapor diffusion in capillaries using deuterated solutions of 2.3 M $KD_2PO_4$ and $K_2HPO_4$ at pH 7.4 (adjusted by varying ratios of dibasic and monobasic forms) that is the equivalent pD value of 7.8. The pD value was calculated from pD = pH$_a$ (apparent reading from pH meter) + 0.4. A crystal in a quartz capillary was soaked in deuterated solutions of 2.3 M $KD_2PO_4/K_2HPO_4$ at pD 7.8 (measured pH 7.4) containing either 6.4 mM potassium permanganate ($KMnO_4$) to achieve the Mn$^{3+}$ state or 300 mM sodium dithionite ($Na_2S_2O_4$) to achieve the Mn$^{2+}$ state. After drying the crystal from soaking solutions, the crystal was flanked in the capillary by slugs of the deuterated reservoir soaking solutions. Fortuitously, the decomposition products of the redox agents are unable to enter the active site of MnSOD[18]. Methods for manipulating the Mn metal of MnSOD to either Mn$^{3+}$ or Mn$^{2+}$ are described in more detail in previous work[18].

**Neutron and X-ray data collection**. Data collection was preceded by the replacement of the deuterated and redox-agent containing reservoir slugs with fresh equivalents. Time-of-flight, wavelength-resolved neutron Laue diffraction data were used to collect data from the 0.46 mm$^3$ perdeuterated crystal using the MaNDi instrument[60,61] at the ORNL SNS using all neutrons with wavelengths between 2-4 Å. Data collection of each diffraction pattern was from the crystal held in a stationary position, with successive diffraction patterns being collected after 20° rotations along the Φ axis. A $KMnO_4$-treated perdeuterated crystal of 0.46 mm$^3$ in volume at 296 K was recorded to 2.20 Å resolution for the Mn$^{3+}$SOD form and subsequently treated with $Na_2S_2O_4$ to achieve the Mn$^{2+}$SOD state where 2.30 Å data were collected (Supplementary Table 6). $Na_2S_2O_4$ is noted to deteriorate diffraction quality and was observed to increase the *c* unit cell axis by ~1 Å[18]. After neutron data were collected from the crystal in the Mn$^{2+}$SOD state, X-ray diffraction data were collected at 296 K to 2.16 Å resolution using a Rigaku FR-E SuperBright home source. After room temperature data collection, the crystal was not suitable for Mn$^{3+}$SOD data collection and a sister crystal grown from the same well was used instead for obtaining X-ray data to 1.87 Å resolution.

**Data processing and refinement**. Neutron data were integrated with MANTID[62] utilizing the three-dimensional profile fitting algorithm[63]. Integrated neutron data were scaled and wavelength-normalized using Lauenorm from the LAUGEN suite[64]. X-ray diffraction data were reduced with HKL-3000 for indexing, integration, and scaling[65]. The refinement of both X-ray and neutron models were completed separately with PHENIX.REFINE from the PHENIX suite[66]. The refinements were intentionally done separately due to the known perturbations that X-rays have on solvent structure and metal redox states which are not present with neutrons[17,67]. The X-ray model was first refined against its corresponding data set and subsequently used as the starting model for neutron refinement. Torsional backbone angle restraints were derived from the X-ray model and applied to neutron refinement using a geometric target function with PHENIX.REFINE[66]. The neutron refinement process was performed to model the D atoms of the active site last to limit phase bias. For the initial rounds of refinement to fit protein structure, only non-exchangeable D atoms (which have stereochemical predictable positions) were present. Afterwards, each individual exchangeable position outside the active site was inspected for residual $|F_o|-|F_c|$ neutron scattering length density and modeled with D atoms appropriately before more iterations of refinement. Next, the O atoms of solvent molecules were first modeled manually outside the active site and refined before determining whether to model solvent as O, OD, or DOD using residual $|F_o|-|F_c|$ neutron scattering length density. The omit density peaks were also used to discern the appropriate orientation of the solvent molecules. After refinement of the solvent structure outside the active site, non-D atoms of the active site were modeled, including Mn and the O of solvent. Last, D atoms of the active site were modeled and refined manually.

**Computational details**. Computational methods are discussed with further depth in the Supplementary Methods. All QM DFT calculations were performed with the NWChem 6.8 software[68]. The COSMO solvation model was implemented into the

geometry optimizations to model the solution phase and utilized an energy convergence value of 1E−6 atomic units[69]. The def2-TZVPD basis set was used for the Mn ion whereas the 6−31+G(d,p) Pople basis set was specifically used for all other atoms due to its use in predicting p$K_a$s under the B3LYP functional[70,71]. The QM models utilized for DFT calculations encompassed the active site residues that had the O and N atoms of the peptide backbone truncated and the C$^\alpha$ atoms fixed. Additional fixed restraints were placed on aromatic residues found on the periphery of the active site (Phe66, Trp123, Trp161, and Tyr166) to mimic the packing found in the native enzyme. The Mn ion used the high-spin quintet and sextet states for trivalent and divalent systems, respectively, per experimental observations[72].

**Bonding orbital analysis**. The JANPA software package was used to calculate CLPOs from open-shell DFT geometry optimizations[37,38,73,74]. These are bonding and antibonding orbitals with maximum electron density computed through a series of localized basis set transformations. CLPOs are calculated with the same target quantity as Natural Bond Orbital (NBO) methods and yield comparable results[35,36,75]. The electron delocalization stabilization/destabilization energies utilized second-order perturbation theory analysis of the Fock matrix defined by the NBO methods[76] but were done in the CLPO basis of JANPA. The energy associated with electron delocalization from lone pair or bonding orbital $i$ to antibonding orbital $j$ is defined in Eq. (1) where $q_i$ is the donor orbital occupancy and $\hat{F}$ is the effective orbital Hamiltonian.[76] Values $\langle i|\hat{F}|j\rangle$ and $\langle i|\hat{F}|i\rangle$ are diagonal CLPO Fock matrix elements indicative of orbital energies and $\langle i|\hat{F}|j\rangle$ is the off-diagonal matrix element representative of perturbation.

$$\Delta E_{i\to j}^2 = -q_i \frac{\langle i|\hat{F}|j\rangle^2}{\langle j|\hat{F}|j\rangle - \langle i|\hat{F}|i\rangle} \tag{1}$$

## Data availability

Coordinates and structure factors for oxidized and reduced MnSOD determined with neutron and X-ray crystallography have been deposited in the Protein Data Bank (PDB 7KKS [https://doi.org/10.2210/pdb7KKS/pdb], 7KKW [https://doi.org/10.2210/pdb7KKW/pdb], 7KKU [https://doi.org/10.2210/pdb7KKU/pdb], and 7KLB [https://doi.org/10.2210/pdb7KLB/pdb]). All relevant data supporting the key findings of this study are available within the article and its Supplementary Information files or from the corresponding author upon reasonable request.

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

## Acknowledgements

We would like to thank Dr. Tymofii Y. Nikolaienko for his support in orbital analysis and utilization of the JANPA software as well as Jeffrey L. Lovelace for his computational aid. We are thankful for the insight obtained from Dr. Andrey Kovalevsky's expertise of inorganic and quantum chemistry as well as his time spent facilitating valuable discussions. We are grateful for the constructive feedback and advice given by Dr. Timothy C. Mueser and Dr. Joseph D. Ng. We thank Scott R. Trickel and Carol Kolar for useful discussions and technical assistance. This research was supported by NASA EPSCoR (NE-80NSSC17M0030 and NE-NNX15AM82A). The UNMC Structural Biology Core Facility was funded by the Fred and Pamela Buffett NCI Cancer Center Support Grant (P30CA036727). L.C. acknowledges support by the NIH (R01-GM071939). The research at Oak Ridge National Laboratory (ORNL) Spallation Neutron Source was sponsored by the Scientific User Facilities Division, Office of Basic Energy Sciences, US Department of Energy. This research used resources at the Second Target Station, which is a DOE Office of Science User Facilities Construction Project at ORNL. The Office of Biological and Environmental Research supported research at ORNL Center for Structural Molecular Biology (CSMB) using facilities supported by the Scientific User Facilities Division, Office of Basic Energy Sciences, US Department of Energy. Quantum computations were completed utilizing the Holland Computing Center of the University of Nebraska, which receives support from the Nebraska Research Initiative.

## Author contributions

J.A., W.E.L., L.C., K.L.W., and G.E.O.B. performed experiments, and analyzed data. J.A. and G.E.O.B wrote the manuscript with input from all authors.

## Competing interests

The authors declare no competing interests.
