## [Peer Review File · Nature Communications]

REVIEWER COMMENTS

Reviewer #1 (Remarks to the Author):

Azadmanesh et al. have collected neutron diffraction data on the reduced and oxidized forms of manganese SOD in an attempt to ascertain the protonation states of residues around the active sites in both oxidation states. Neutron diffraction is not frequently used on enzymatic samples, and I consequently find the study to be quite novel on that basis alone.

The results are highly intriguing and provide insight into how previously studied mutations altered the activity of the enzyme. Unexpectedly, Gln143 appears to be deprotonated in the reduced form of the enzyme. This is highly counter-intuitive since Mn(II)-OH₂ groups would normally be more acidic than amides. To their credit, the authors are clearly cognizant of this oddity and provide a clear rationale for why the amide of Gln143 may be more acidic than anticipated.

My only substantial concern is that the authors' methodology likewise suggests that His 30 may exist as an anionic imidazolate, which is chemically implausible. Could the anionic Gln143 residue likewise be an artifact?

Once this issue is addressed, the manuscript should be suitable for publication in Nature Communications.

A. On page 6, the authors describe the interaction between Mn(II) and water as being between a soft acid and a soft base. Although these are softer than Mn(III) and hydroxide, respectively, both are still considered to be hard on the HSAB scale.

B. I am concerned that the data also suggest a chemically unrealistic imidazolate anion for His30. I am glad that the authors discount this possibility, but I worry that similarly misleading data may be suggesting an artificial amidate for Gln143.

C. I had trouble following the proton and electron relays in Figure 6. An accompanying Chemdraw scheme might elucidate matters.

Reviewer #2 (Remarks to the Author):

In this manuscript, the authors present neutron structures of Mn superoxide dismutase for both the Mn(II) and Mn(III) states. The structures show several very interesting features: The Mn(II) state, shows a OH₂ ligand, as expected, but the proton is delivered from the -NH₂ group of Gln143, which becomes deprotonated. In one of the two active sites in the protein, Mn(II) becomes 6-coordinate with two OH⁻ ligands. In the oxidized state in one of the two active sites, His30 is a deprotonated imidazolate group. All these are very interesting suggestions and merit publication. However, a few clarifications are needed first.

1. Several mechanisms for MnSOD have been suggested based on QM calculations. They should be discussed.
2. Some discussion of previous crystallographic studies of MnSOD should also be mentioned in the introduction.
3. The study in ref. 17 actually is an experimental (X-ray) observation of protonated water in the Mn(II) state.
4. From an inorganic perspective, it seems quite unlikely that Mn(II) should have a single H₂O ligand in one subunit but suddenly two OH⁻ ligands in the other subunit.
5. The authors should better explain what they mean by differential protonation and the difference between Fig 5a and 5b.

6. Fig 5b should show the omit difference density of the proton on Wat2 also.
7. To put the reported H-bond covalencies in a perspective, the authors should report the covalency of a normal H₂O–HOH H-bond, calculated with the same QM method.
8. The suggested reaction mechanism in 6 is a bit strange as it ignores the substrates.
9. Na₂S₂O₄ is not sodium hydrosulfite.
10. The type of refinement is not fully clear. Did you perform joint refinement of the neutron and X-ray data?
11. I do not think you used COSMO-RS in the QM calculations, only COSMO.
12. A convergence criterium of 5E-4 a.u. is very floppy. Most QM programs have closer to 1E-6 a.u., i.e. 500 times stricter.
13. The difference between Figures S3a and b should be clarified.

Reviewer #3 (Remarks to the Author):

The mechanisms of the concerted proton and electron transfers (CEPTs) of human manganese superoxide dismutase (MnSOD) are provided in the paper. Since authors' most important statement, "a novel mechanism is proposed from the direct observation of glutamine deprotonation, the involvement of Tyr and His with altered pKaS, and the three unusual strong-short hydrogen bonds that change with the oxidation state of the metal Mn", is strongly based on the neutron protein crystallography (NPC) results, the accuracy of the NPC results is decisive. Therefore, there are several questions about the NPC results. When these would be solved, we can admit that NPC could contribute to detect the CEPTs of MnSOD well.

1) When discussing the pKa of amino acids, the pH value of the sample is very important. It seems that crystallization was performed at physiological pH, but it is necessary to describe the pH (or pD) value of the protein solution under the crystallization. What was the actual pH(or pD) in both oxidized and reduced cases?

2) There are two subunits in the crystallographic asymmetric unit. In case of the reduced state, did the structure shown in Figure 2b and the structure shown in Figure 4c coexist? If so, why? And, in case of the oxidized state does only the structure shown in Figure 2a exist? If so, why were there such differences between the two states?

3) Why the Dε22 (Q143) cannot be seen in the Figure 2a ?

4) Line 108-116: Since the Fourier map peak of D 2 (WAT1) is rather small, it is dangerous to conclude that this does not belong to the Dε21 (Q143). The authors have concluded that O(WAT1)-D 2 (WAT1)-Nε2(Gln143) is a short-strong hydrogen bond (SSHB). The criteria whether it is a SSHB or not are that (1)the difference of pKa between donor and acceptor atoms should be near zero, (2)the hydrogen bond distance is short, and (3)the hydrogen atom is located roughly equidistant from the donor and acceptor atoms. O(WAT1)-D 2 (WAT1)-Nε2(Gln143) is a simple hydrogen bond, isn't it? It is necessary to consider whether the other SSHBs appeared in the paper also fit the above mentioned criteria.

5) Why is the Dε1(W123) in Figure 2b rather smaller than Dε1(W123) in Figure 2a?

6) Line 285-286: It is a little bit dangerous to conclude His30 has an imidazolate anion. It is often observed that a histidine on the surface of a protein looks like an imidazolate anion simply because the B-factor of a hydrogen atom is too large and the hydrogen atom cannot be seen on the Fourier map. The B-factor of the Dδ 1 is too large and it cannot be seen on the map, isn't it?

7) The Authors use the word "nuclear density" for the peak on the Fourier contour map so often. It's a matter of technical terms, and strictly speaking, they should use a "neutron scattering length (NSL) density", but not a "nuclear density". The unit of the neutron scattering power is "neutron scattering length" and the peak on the Fourier contour map obtained by neutron diffraction Bragg reflections should be called as "neutron scattering length (NSL) density", but not "nuclear density". For example, the neutron scattering length of Mn is negative, such as -0.373×10^{-12} cm. So, the peak of Mn on the Fourier contour map must be obtained as negative value. The "negative nuclear density" has physically no sense, but "negative NSL density" is just OK.

Places in the manuscript that were changed are highlighted in light grey.

Reviewer #1 (Remarks to the Author):

Review of Manuscript NCOMMS-20-43515

Azadmanesh et al. have collected neutron diffraction data on the reduced and oxidized forms of manganese SOD in an attempt to ascertain the protonation states of residues around the active sites in both oxidation states. Neutron diffraction is not frequently used on enzymatic samples, and I consequently find the study to be quite novel on that basis alone.

The results are highly intriguing and provide insight into how previously studied mutations altered the activity of the enzyme. Unexpectedly, Gln143 appears to be deprotonated in the reduced form of the enzyme. This is highly counter-intuitive since Mn(II)-OH₂ groups would normally be more acidic than amides. To their credit, the authors are clearly cognizant of this oddity and provide a clear rationale for why the amide of Gln143 may be more acidic than anticipated.

My only substantial concern is that the authors' methodology likewise suggests that His 30 may exist as an anionic imidazolate, which is chemically implausible. Could the anionic Gln143 residue likewise be an artifact?

Once this issue is addressed, the manuscript should be suitable for publication in Nature Communications.

A. On page 6, the authors describe the interaction between Mn(II) and water as being between a soft acid and a soft base. Although these are softer than Mn(III) and hydroxide, respectively, both are still considered to be hard on the HSAB scale. **The reviewer is correct in that Mn(II) and water are considered hard on the HSAB scale and our wording was technically incorrect. We have rewritten this portion of the manuscript.**

B. I am concerned that the data also suggest a chemically unrealistic imidazolate anion for His30. I am glad that the authors discount this possibility, but I worry that similarly misleading data may be suggesting an artificial amidate for Gln143. **We have completely reanalyzed our data concerning His30 and have included more data into the manuscript based on our new observations. Reviewer #1 comments (and reviewer #3 below) helped us to look at the data with a new critical eye. We have refocused this section on the low barrier hydrogen bond that exists between His30 and Tyr166 across the dimer interface in the reduced state and the apparent altered pKas of this pair of amino acids that is probably important to the enzymatic mechanism. We also included a B-value table in the supplement (Table S4) to help the reader evaluate the data for His 30 in the oxidized state, if they are interested.**

The data concerning Gln143 is without question. Gln143 was identified as an amide anion in the apo MnSOD reduced state due to the presence of the neutron scattering length density. While the peak is indeed small relative to other peaks at the same contour it would not be unexpected if there is indeed movement of the proton between WAT1 and Q143. Simply lowering the contour to 2.3 σ from 2.5 σ puts the density size close to the other 2.5 σ peaks. This has been added to the figure. We realize that we are suggesting unlikely chemistry for the Q143-WAT1 interaction and thus sought validation through QM calculations as demonstrated with Figure 3 and careful investigation of the literature. We believe the culmination of the experimental data, the QM calculations, and past studies noted in the manuscript thus strongly favor presence of the amide anion over the neutral amide.

C. I had trouble following the proton and electron relays in Figure 6. An accompanying Chemdraw scheme might elucidate matters. **We have included Chemdraw summaries of the results as well as revised the final suggested mechanism figure into a Chemdraw scheme. We find this to be a great improvement in clarity, thank you for the suggestion.**

Reviewer #2 (Remarks to the Author):

In this manuscript, the authors present neutron structures of Mn superoxide dismutase for both the Mn(II) and Mn(III) states. The structures show several very interesting features: The Mn(II) state, shows a OH₂ ligand, as expected, but the proton is delivered from the -NH₂ group of Gln143, which becomes deprotonated. In one of the two active sites in the protein, Mn(II) becomes 6-coordinate with two OH⁻ ligands. In the oxidized state in one of the two active sites, His30 is a deprotonated imidazolate group. All these are very interesting suggestions and merit publication. However, a few clarifications are needed first.

1. Several mechanisms for MnSOD have been suggested based on QM calculations. They should be discussed.
2. Some discussion of previous crystallographic studies of MnSOD should also be mentioned in the introduction. For suggestions 1 & 2 from reviewer #2, a paragraph has been added into the introduction that discusses previous crystallographic and QM studies.
3. The study in ref. 17 actually is an experimental (X-ray) observation of protonated water in the Mn(II) state. This is correct and the text of the manuscript has been edited accordingly.
4. From an inorganic perspective, it seems quite unlikely that Mn(II) should have a single H₂O ligand in one subunit but suddenly two OH⁻ ligands in the other subunit. We agree with Reviewer #2 and also consider Mn(II) with two OH⁻ ligands as implausible. However, the respective active site is not strictly two OH⁻ ligands, but rather one strict OH⁻ ligand and another OH⁻ ligand with partial HOH character from its strong short hydrogen bond (1.4 Å) with the proximal amide proton. This is further supported by the intermediary Mn-O(WAT1) bond distance of 2.1 Å compared to the 2.2 Å Mn-O(WAT1) bond distance of the opposing chain where the data indicates H₂O as the identity of WAT1. X-ray structures of MnSOD are known to differ in coordination between chains of the asymmetric unit [1-2].

We also performed DFT geometry optimizations to assess the plausibility of the six-coordinate Mn(II) seen with the neutron data. Indeed, it is stable and has Mn-O(WAT1) bond distance of 2.1 Å. As a further test, replacing the ⁻OH ligand with H₂O at the sixth coordinate position causes the H₂O molecule to disassociate from Mn(II) during DFT geometry optimizations. These details have been added to the manuscript.

[1] Borgstahl, G. E. O., Pokross, M., Chehab, R., Sekher, A. & Snell, E. H. Cryo-trapping the six-coordinate, distorted-octahedral active site of manganese superoxide dismutase. *J. Mol. Biol.* **296**, 951-959, doi:10.1006/jmbi.1999.3506 (2000).

[2] Porta, J., Vahedi-Faridi, A. & Borgstahl, G. E. O. Structural analysis of peroxide-soaked MnSOD crystals reveals side-on binding of peroxide to active-site manganese. *J. Mol. Biol.* **399**, 377-384, doi:10.1016/j.jmb.2010.04.031 (2010).

5. The authors should better explain what they mean by differential protonation and the difference between Fig 5a and 5b. We understand the confusion of the figures. We have rewritten the text to be more explicit in the changes of protonation.
6. Fig 5b should show the omit difference density of the proton on Wat2 also. Good catch, this has been added.
7. To put the reported H-bond covalencies in a perspective, the authors should report the covalency of a normal H₂O–HOH H-bond, calculated with the same QM method. This is a good suggestion and we have implemented this value into the text, ~90/10 covalence from the same QM method.
8. The suggested reaction mechanism in 6 is a bit strange as it ignores the substrates. We agree that this is atypical and may be off-putting. However, we do not have experimental data with the substrates (or products) that may change the protonation states of the active site – though it is certainly in the works. The benefit with the present work is the changes of protonation states that occur not as a consequence of the substrate but rather

the redox state.

9. Na₂S₂O₄ is not sodium hydrosulfite. Corrected

10. The type of refinement is not fully clear. Did you perform joint refinement of the neutron and X-ray data? The final stages of refinement were intentionally done separately from the X-ray data due to the lack of isomorphism caused by the known perturbations that X-rays have on solvent structure and metal redox states which are not present with neutrons. X-rays are substantially reducing to metalloproteins and consequentially make it difficult to obtain a totally oxidized structure. Also, X-rays are known to perturb ordered solvent which are of importance for studying MnSOD. The neutron refinement implemented restraints from the refined X-ray structure with the PHENIX refinement software. These details have been clarified in the methods section along with appropriate references.

11. I do not think you used COSMO-RS in the QM calculations, only COSMO. The reviewer is correct and the methods sections have been corrected. The appropriate reference has also been added.

12. A convergence criterium of 5E-4 a.u. is very floppy. Most QM programs have closer to 1E-6 a.u., i.e. 500 times stricter. This was an error and in reality the expected value had been used. The orbital gradient convergence criterion was mistakenly interpreted as the energy convergence criterion. The version of the NWChem QM software used indeed has a default energy convergence criterion of 1E-6 that was unaltered during calculations. See images below. This has been clarified in both the methods section and the supplementary methods section.

```
Convergence on energy requested: 1.00D-06
Convergence on density requested: 1.00D-05
Convergence on gradient requested: 5.00D-04
```

Convergence utilized as noted from an output file.

- convergence of the total energy; this is defined to be when the total DFT energy at iteration N and at iteration N-1 differ by a value less than some value (the default is 1e-6). This value can be modified using the key word,

```
CONVERGENCE energy <real energy default 1e-6>
```

Documentation from NWChem denoting the default energy convergence value.
<https://nwchemgit.github.io/Density-Functional-Theory-for-Molecules.html>

13. The difference between Figures S3a and b should be clarified. Figures S3a-b have been reworked into the main manuscript into Figure 5 and differences have been clarified, specifically the chain identity is identified and has a color assigned to it.

Reviewer #3 (Remarks to the Author):

The mechanisms of the concerted proton and electron transfers (CEPTs) of human manganese superoxide dismutase (MnSOD) are provided in the paper. Since authors' most important statement, "a novel mechanism is proposed from the direct observation of glutamine deprotonation, the involvement of Tyr and His with altered pKaS, and the three unusual strong-short hydrogen bonds that change with the oxidation state of the metal Mn", is strongly based on the neutron protein crystallography (NPC) results, the accuracy of the NPC results is decisive. Therefore, there are several questions about the NPC results. When these would be solved, we can admit that NPC could contribute to detect the CEPTs of MnSOD well.

1) When discussing the pK_a of amino acids, the pH value of the sample is very important. It seems that crystallization was performed at physiological pH, but it is necessary to describe the pH (or pD) value of the protein solution under the crystallization. What was the actual pH(or pD) in both oxidized and reduced cases?

We understand the lack of clarity here and revised the text of the methods section to be explicit about the pH/pD both during crystallization and data collection.

2) There are two subunits in the crystallographic asymmetric unit. In case of the reduced state, did the structure shown in Figure 2b and the structure shown in Figure 4c coexist? If so, why? And, in case of the oxidized state does only the structure shown in Figure 2a exist? If so, why were there such differences between the two states?

We understand how the topic of chain identity may be puzzling to the reader and have reworked the manuscript to specify chain identity. For example, note chain labels in Fig 1a corresponding to the color of the chain.

There were differences between 2b and 4c. We attribute this to differences in solvent accessibility as noted in supplementary figure S3. Previous MnSOD structures are known to have differences in coordination among the chains [1-2]. For the oxidized state, Fig. 2a, both subunits looked the same. These have been clarified.

[1] Borgstahl, G. E. O., Pokross, M., Chehab, R., Sekher, A. & Snell, E. H. Cryo-trapping the six-coordinate, distorted-octahedral active site of manganese superoxide dismutase. *J. Mol. Biol.* **296**, 951-959, doi:10.1006/jmbi.1999.3506 (2000).

[2] Porta, J., Vahedi-Faridi, A. & Borgstahl, G. E. O. Structural analysis of peroxide-soaked MnSOD crystals reveals side-on binding of peroxide to active-site manganese. *J. Mol. Biol.* **399**, 377-384, doi:10.1016/j.jmb.2010.04.031 (2010).

3) Why the Dε22 (Q143) cannot be seen in the Figure 2a ? **This has been added!**

4) Line 108-116: Since the Fourier map peak of D 2 (WAT1) is rather small, it is dangerous to conclude that this does not belong to the Dε21 (Q143). The authors have concluded that O(WAT1)-D 2 (WAT1)-Nε2(Gln143) is a short-strong hydrogen bond (SSHB). The criteria whether it is a SSHB or not are that (1)the difference of pK_a between donor and acceptor atoms should be near zero, (2)the hydrogen bond distance is short, and (3)the hydrogen atom is located roughly equidistant from the donor and acceptor atoms. O(WAT1)-D 2 (WAT1)-Nε2(Gln143) is a simple hydrogen bond , isn't it? It is necessary to consider whether the other SSHBs appeared in the paper also fit the above mentioned criteria. **While the peak is indeed small relative to other peaks at the same contour it would not be unexpected if there is indeed movement of the proton between WAT1 and Q143. Simply lowering the contour to 2.3 σ from 2.5 σ puts the density size close to the other 2.5 σ peaks. This has been added to the figure. Likewise, one would expect a larger peak if the proton belonged to Q143 and acted as a hydrogen bond donor to WAT1. We realize that we are suggesting unlikely chemistry for the Q143-WAT1 interaction and thus sought validation through QM calculations as demonstrated with Figure 3 and careful investigation of the literature. We believe the culmination of the experimental data, the QM calculations, and past studies noted in the manuscript thus strongly favor presence of the amide anion over the neutral amide.**

Investigation of the literature suggests that the term SSHB is sometimes used interchangeably with the term 'low-barrier hydrogen bond' (LBHB) [1] and thus generates confusion. Also, there seems to be different definitions of the terms among studies. To remediate this, we have decided to explicitly define these terms in the manuscript.

A SSHB is simply a shorter hydrogen bond that indicates greater strength that may have enzymatic implications [2]. For example, the His, Asp, and Ser of catalytic triads have short hydrogen bonds and stark differences of canonical pK_a values but use proton transfers.

A LBHB is a type of SSHB with the added characteristics of (1) the difference of pKa between donor and acceptor atoms should be near zero and (2) the hydrogen atom is located roughly equidistant from the donor and acceptor atoms [3]. This is the same definition used by a recent 2019 Nature publication [4].

[1] Cleland WW. Low-barrier hydrogen bonds and enzymatic catalysis. *Arch Biochem Biophys.* 2000 Oct 1;382(1):1-5. doi: 10.1006/abbi.2000.2011. PMID: 11051090.

[2] Remer LC., and Jensen JH. Toward a general theory of hydrogen bond:the short, strong hydrogen bond. *J. Phys. Chem. A* 2000, 104, 40, 9266–9275. doi:10.1021/jp002726n

[3] Perrin, C. L. & Nielson, J. B. “Strong” hydrogen bonds in chemistry and biology. *Annu. Rev. Phys. Chem.* **48**, 511–544 (1997).

[4] Dai, S., Funk, LM., von Pappenheim, F.R. *et al.* Low-barrier hydrogen bonds in enzyme cooperativity. *Nature* **573**, 609–613 (2019). <https://doi.org/10.1038/s41586-019-1581-9>

5) Why is the $D\epsilon 1(W123)$ in Figure 2b rather smaller than $D\epsilon 1(W123)$ in Figure 2a? While the neutron data sets from the reduced state (2b) and the oxidized state (2a) are indeed from the same crystal, the strength of the Fourier map peaks should not be expected to be equal. There are notable differences in data collection statistics that may contribute to the difference in Fourier map peaks. The most apparent examples are differences among the unit cell dimensions, resolution, and $I/\sigma(I)$ as seen in table S5.

6) Line 285-286: It is a little bit dangerous to conclude His30 has an imidazolate anion. It is often observed that a histidine on the surface of a protein looks like an imidazolate anion simply because the B-factor of a hydrogen atom is too large and the hydrogen atom cannot be seen on the Fourier map. The B-factor of the $D\delta 1$ is too large and it cannot be seen on the map, isn't it? The section for His30 has been reworked and does not emphasize the possibility of an imidazolate anion. The section was refocused on the LBHB between His30 and Tyr166 across the dimer interface.

7) The Authors use the word “nuclear density” for the peak on the Fourier contour map so often. It's a matter of technical terms, and strictly speaking, they should use a “neutron scattering length (NSL) density”, but not a “nuclear density”. The unit of the neutron scattering power is “neutron scattering length” and the peak on the Fourier contour map obtained by neutron diffraction Bragg reflections should be called as “neutron scattering length (NSL) density”, but not “nuclear density”. For example, the neutron scattering length of Mn is negative, such as -0.373×10^{-12} cm. So, the peak of Mn on the Fourier contour map must be obtained as negative value. The “negative nuclear density” has physically no sense, but “negative NSL density” is just OK. We understand this issue of technical terms and wish for the manuscript to be the most technically sound as possible. We have edited the manuscript accordingly.

REVIEWER COMMENTS

Reviewer #1 (Remarks to the Author):

The authors have adequately addressed my concerns. In my opinion, the manuscript can be published as is.

Reviewer #2 (Remarks to the Author):

This is the revised version of a manuscript describing neutron structures of Mn superoxide dismutase for both the Mn(II) and Mn(III) states. The authors have made strong improvements and the manuscript can now be accepted after two very minor points have been clarified.

1. The text says that the H-bond length between Asp159 and Wat1 is 2.1 Å, but Figure 2a says 1.9 Å.

Line 272: "suggests stability of the six-coordinate complex": What six-coordinate complex? What ox state and what ligands?

Reviewer #3 (Remarks to the Author):

I have re-reviewed the revised version mainly according to my previous questions. I found there still were unclear parts as follows: (Italic: comments and new questions)

1) When discussing the pKa of amino acids, the pH value of the sample is very important. It seems that crystallization was performed at physiological pH, but it is necessary to describe the pH (or pD) value of the protein solution under the crystallization. What was the actual pH(or pD) in both oxidized and reduced cases? ----◇ Line 443-448: Normally pD is determined as pD is added 0.4 to the measured pH value with a pH meter. According to the revised version, both pH and pD are the same value. Have you adjusted as pD is the same as pH? How have you done it and was it necessary to adjust that pD and pH are the same value?

2) There are two subunits in the crystallographic asymmetric unit. In case of the reduced state, did the structure shown in Figure 2b and the structure shown in Figure 4c coexist? If so, why? And, in case of the oxidized state does only the structure shown in Figure 2a exist? If so, why were there such differences between the two states? --◇ I have understood that in one crystallographic asymmetric unit there are two chains, chain A and chain B. Moreover, in the reduced, the protonation structures between chain A and chain B are not the same. How is the case of oxidized state between chain A and chain B? Is it OK to discuss the CEPTs by using together the structure of chain A and chain B, but not using independently only chain A or chain B ?

3) Why the Dε22 (Q143) cannot be seen in the Figure 2a ? --◇ (OK)

4) Line 108-116: Since the Fourier map peak of D 2 (WAT1) is rather small, it is dangerous to conclude that this does not belong to the Dε21 (Q143). The authors have concluded that O(WAT1)-D 2 (WAT1)-Nε2(Gln143) is a short-strong hydrogen bond (SSHB). The criteria whether it is a SSHB or not are that (1)the difference of pKa between donor and acceptor atoms should be near zero, (2)the hydrogen bond distance is short, and (3)the hydrogen atom is located roughly equidistant from the donor and acceptor atoms. O(WAT1)-D 2 (WAT1)-Nε2(Gln143) is a simple hydrogen bond , isn't it? It is necessary to consider whether the other SSHBs appeared in the paper also fit the above mentioned criteria. --◇ (OK)

- 5) Why is the $D_{\epsilon 1}(W123)$ in Figure 2b rather smaller than $D_{\epsilon 1}(W123)$ in Figure 2a?---◇ (OK)
- 6) Line 285-286: It is a little bit dangerous to conclude His30 has an imidazolate anion. It is often observed that a histidine on the surface of a protein looks like an imidazolate anion simply because the B-factor of a hydrogen atom is too large and the hydrogen atom cannot be seen on the Fourier map. The B-factor of the $D_{\delta 1}$ is too large and it cannot be seen on the map, isn't it? --◇(OK)
- 7) The Authors use the word "nuclear density" for the peak on the Fourier contour map so often. It's a matter of technical terms, and strictly speaking, they should use a "neutron scattering length (NSL) density", but not a "nuclear density". The unit of the neutron scattering power is "neutron scattering length" and the peak on the Fourier contour map obtained by neutron diffraction Bragg reflections should be called as "neutron scattering length (NSL) density", but not "nuclear density". For example, the neutron scattering length of Mn is negative, such as -0.373×10^{-12} cm. So, the peak of Mn on the Fourier contour map must be obtained as negative value. The "negative nuclear density" has physically no sense, but "negative NSL density" is just OK. --◇ (line 85-86) The sentence 'Mn scatters negatively and therefore lacks density' has no sense. Simply, the neutron scattering length of Mn is negative.

Places in the manuscript that were changed are highlighted in light grey.

Reviewer #1 (Remarks to the Author):

The authors have adequately addressed my concerns. In my opinion, the manuscript can be published as is.

We kindly thank the reviewer for their suggestions that have improved the manuscript.

Reviewer #2 (Remarks to the Author):

This is the revised version of a manuscript describing neutron structures of Mn superoxide dismutase for both the Mn(II) and Mn(III) states. The authors have made strong improvements and the manuscript can now be accepted after two very minor points have been clarified.

1. The text says that the H-bond length between Asp159 and Wat1 is 2.1 Å, but Figure 2a says 1.9 Å.

Line 272: “suggests stability of the six-coordinate complex”: What six-coordinate complex? What ox state and what ligands?

We are thankful to the reviewer for the suggestions in clarification! The text has been corrected to 1.9 Å to correspond with Figure 2a. The sentence of line 272 has been clarified:

>Line 272: Further validation was sought with DFT geometry optimizations of the observed neutron structure, Mn²⁺SOD with sixth-coordinate ⁻OH(OL), and suggests stability of the six-coordinate complex while replacing OL with H₂O causes disassociation into five-coordinate Mn²⁺SOD.

Reviewer #3 (Remarks to the Author):

I have re-reviewed the revised version mainly according to my previous questions. I found there still were unclear parts as follows: (Italic: comments and new questions)

1) When discussing the pK_a of amino acids, the pH value of the sample is very important. It seems that crystallization was performed at physiological pH, but it is necessary to describe the pH (or pD) value of the protein solution under the crystallization. What was the actual pH(or pD) in both oxidized and reduced cases?

---◇ Line 443-448: Normally pD is determined as pD is added 0.4 to the measured pH value with a pH meter. According to the revised version, both pH and pD are the same value. Have you adjusted as pD is the same as pH? How have you done it and was it necessary to adjust that pD and pH are the same value?

We thank the reviewer for pointing out the lack of clarity. We have added the explicit mention of measured pH values that led to the pD values discussed.

>Line 443-445 now reads: Deuterium exchange of crystals was performed by vapor diffusion in capillaries using deuterated solutions at pH 7.4 that is the equivalent pD value of 7.8. The pD value was calculated from pD = pH_a (apparent reading from pH meter) + 0.4.

>Line 448 indicates that the pD value of 7.8 was obtained by adding 0.4 to the measured pH value.

2) There are two subunits in the crystallographic asymmetric unit. In case of the reduced state, did the structure shown in Figure 2b and the structure shown in Figure 4c coexist? If so, why? And, in case of the oxidized state does only the structure shown in Figure 2a exist? If so, why were there such differences between the two states?

---◇ I have understood that in one crystallographic asymmetric unit there are two chains, chain A and chain B. Moreover, in the reduced, the protonation structures between chain A and chain B are not the same. How is the case of oxidized state between chain A and chain B? Is it OK to discuss the CEPTs by using together the structure of chain A and chain B, but not using independently only chain A or chain B ?

The reviewer makes a good point and we have added the following clarifications.

For the oxidized MnSOD, there is high structural similarity between Chain A and Chain B. The only change in protonation between the chains is the protonation state of His30. When discussing the oxidized structure in the conclusions, we have clarified:

>Line 367-371: For the oxidized resting state that is described by both chains of the Mn³⁺SOD neutron structure (Fig. 6a), the proton bridging His30 and Tyr166 appears to be moving and this suggests the possibility of Tyr166 alternating between an ionized or protonated form and a deprotonated or protonated N^{ε2}(His30). N^{δ1}(His30) is also observed to be both deprotonated or protonated (Fig. 5c-d). For simplicity, only one protonation form of His30 and Tyr34 is shown (Fig 6a).

>Line 375: For the reduced resting state described by chain B of Mn²⁺SOD (Fig. 6b),

>Line 379-380: The third active site state is a six-coordinate Mn²⁺ with ⁻OH bound opposite Asp159 (Fig. 6c) described by chain A of Mn²⁺SOD.

Otherwise, we have interpreted the two chains for Mn³⁺SOD as one structural state. This is the reason why we only included figures of one chain for Fig. 2 and Fig. 4 in the manuscript while placing figures of the other chain in the supplemental. For the figures legends, we have clarified:

>Legend 2a: Both panels are for chain B. Only one chain is shown due to high structural similarities, see Fig. S1 for chain A.

>Legend 4a: For the oxidized state, only one chain is shown due to high structural similarities, see Fig. S3 for chain B.

For the reduced MnSOD, the chains have stark differences, especially the coordination state of Mn²⁺. Other differences include protonation changes at Tyr34, Gln143, and WAT1. For these reasons, we have interpreted the chains as being in two different states and is the apparent reason discussion of CPETs considers chain A and chain B two independent structures.

3) Why the Dε22 (Q143) cannot be seen in the Figure 2a ? --◇ (OK)

The reviewer has indicated that this comment was addressed.

4) Line 108-116: Since the Fourier map peak of D2(WAT1) is rather small, it is dangerous to conclude that this does not belong to the Dε21 (Q143). The authors have concluded that O(WAT1)-D2(WAT1)-Nε2(Gln143) is a short-strong hydrogen bond (SSHB). The criteria whether it is a SSHB or not are that (1)the difference of pKa between donor and acceptor atoms should be near zero, (2)the hydrogen bond distance is short, and (3)the hydrogen atom is located roughly equidistant from the donor and acceptor atoms. O(WAT1)-D2(WAT1)-Nε2(Gln143) is a simple hydrogen bond , isn't it? It is necessary to consider whether the other SSHBs appeared in the paper also fit the above mentioned criteria. --◇ (OK)

The reviewer has indicated that this comment was addressed.

5) Why is the Dε1(W123) in Figure 2b rather smaller than Dε1(W123) in Figure 2a?---◇ (OK)

The reviewer has indicated that this comment was addressed.

6) Line 285-286: It is a little bit dangerous to conclude His30 has an imidazolate anion. It is often observed that a histidine on the surface of a protein looks like an imidazolate anion simply because the B-factor of a hydrogen atom is too large and the hydrogen atom cannot be seen on the Fourier map. The B-factor of the Dδ 1 is too large and it cannot be seen on the map, isn't it? --◇(OK)

The reviewer has indicated that this comment was addressed.

7) The Authors use the word “nuclear density” for the peak on the Fourier contour map so often. It's a matter of technical terms, and strictly speaking, they should use a “neutron scattering length (NSL) density”, but not a “nuclear density”. The unit of the neutron scattering power is “neutron scattering length” and the peak on the Fourier contour map obtained by neutron diffraction Bragg reflections should be called as “neutron scattering length (NSL) density”, but not “nuclear density”. For example, the neutron scattering length of Mn is negative, such as -0.373 x 10⁻¹² cm. So, the peak of Mn on the Fourier contour map must be obtained as negative value. The “negative nuclear density” has physically no sense, but “negative NSL density” is just OK.

--◇ (line 85-86) The sentence ‘ Mn scatters negatively and therefore lacks density’ has no sense. Simply, the neutron scattering length of Mn is negative.

The reviewer is correct and we thank them for the assistance in technicalities.

Line 85-85 reads “Of note, the neutron scattering length of Mn is negative.”

REVIEWERS' COMMENTS

Reviewer #3 (Remarks to the Author):

The authors have adequately solved my questions. In my opinion, the manuscript can be published as is.